# Deep cell phenotyping and spatial analysis of multiplexed imaging with TRACERx-PHLEX

Alastair Magness[1,13] ✉, Emma Colliver [1,13], Katey S. S. Enfield [1,13], Claudia Lee [1], Masako Shimato [1], Emer Daly[1], David A. Moore [1,2,3], Monica Sivakumar[2], Karishma Valand [4], Dina Levi[5], Crispin T. Hiley[1,2], Philip S. Hobson[5], Febe van Maldegem [4,10,11,12], James L. Reading [2,6,7], Sergio A. Quezada [2,7], Julian Downward [4], Erik Sahai [8], Charles Swanton [1,2,9] ✉ & Mihaela Angelova [1] ✉

The growing scale and dimensionality of multiplexed imaging require reproducible and comprehensive yet user-friendly computational pipelines. TRACERx-PHLEX performs deep learning-based cell segmentation (deep-imcyto), automated cell-type annotation (TYPEx) and interpretable spatial analysis (Spatial-PHLEX) as three independent but interoperable modules. PHLEX generates single-cell identities, cell densities within tissue compartments, marker positivity calls and spatial metrics such as cellular barrier scores, along with summary graphs and spatial visualisations. PHLEX was developed using imaging mass cytometry (IMC) in the TRACERx study, validated using published Co-detection by indexing (CODEX), IMC and orthogonal data and benchmarked against state-of-the-art approaches. We evaluated its use on different tissue types, tissue fixation conditions, image sizes and antibody panels. As PHLEX is an automated and containerised Nextflow pipeline, manual assessment, programming skills or pathology expertise are not essential. PHLEX offers an end-to-end solution in a growing field of highly multiplexed data and provides clinically relevant insights.

High-dimensional tissue imaging has enabled detailed cell phenotyping in situ, which is essential to understanding cell function and coordination. As the number of simultaneously profiled markers and samples grows, data analysis and interpretation increase in complexity. Multiplexed image analysis typically includes five integral steps: image preprocessing, cell segmentation, cell phenotyping, spatial analysis and visualisation.

End-to-end solutions for multiplexed imaging include collections of tools[1,2] and Nextflow pipelines[3], and often require substantial manual intervention. While automated image preprocessing and deep

[1]Cancer Evolution and Genome Instability Laboratory, The Francis Crick Institute, London, UK. [2]Cancer Research UK Lung Cancer Centre of Excellence, University College London Cancer Institute, London, UK. [3]Department of Cellular Pathology, University College London Hospitals, London, UK. [4]Oncogene Biology Laboratory, The Francis Crick Institute, London, UK. [5]Flow Cytometry, The Francis Crick Institute, London, UK. [6]Pre-cancer Immunology Laboratory, University College London Cancer Institute, London, UK. [7]Immune Regulation and Tumour Immunotherapy Group, Cancer Immunity Unit, Research, Department of Haematology, University College London Cancer Institute, London, UK. [8]Tumour Cell Biology Laboratory, The Francis Crick Institute, London, UK. [9]Department of Oncology, University College London Hospitals, London, UK. [10]Present address: Department of Molecular Cell Biology and Immunology, Amsterdam UMC, Location VUMC, Amsterdam, The Netherlands. [11]Present address: Cancer Center Amsterdam, Cancer Biology and Immunology, Amsterdam, The Netherlands. [12]Present address: Amsterdam Institute for Infection and Immunity, Cancer Immunology, Amsterdam, The Netherlands. [13]These authors contributed equally: Alastair Magness, Emma Colliver, Katey S. S. Enfield. ✉e-mail: alastair.magness@crick.ac.uk; charles.swanton@crick.ac.uk; mihaela.angelova@crick.ac.uk

learning-based segmentation approaches have been integrated into modularly designed Nextflow pipelines for multiplexed imaging[4], there remains a need for user-friendly automated pipelines geared toward imaging mass cytometry (IMC) with image preprocessing steps, larger ground truth datasets and pretrained segmentation models. In addition, currently available pipelines typically use unsupervised clustering approaches for cell subtype identification[3,4], which involve cluster adjustments and manual annotation based on average marker expression.

Recently, standalone probabilistic approaches have enabled an automated identification and annotation of cell subtypes using multiplexed imaging, given a user-provided list of cell subtype-specific markers and a threshold for assignment probabilities[5,6]. While deep learning approaches are not limited to a priori subtype definitions, training data for a given antibody panel may be required[7]. Therefore, fully automated approaches that comprehensively identify both cell subtypes and positivity of functional markers are needed.

To address these needs, we developed TRACERx-PHLEX (cell PHenotype and Localisation analysis of multiplEXed imaging), a comprehensive yet user-friendly pipeline for multiplexed imaging analysis and interpretation. As a modular Nextflow pipeline, PHLEX first incorporates a deep learning-based segmentation model with a ground truth dataset for IMC. Second, it automates the complex task of cell phenotyping through cell subtype identification, assignment and marker positivity calling. Third, PHLEX incorporates a suite of spatial analysis methods facilitating measurement of clinically relevant tissue architectural features, such as the extended occlusion of tumour cells by stromal cells[8].

The visualisations, graphs and tables automatically output by PHLEX allow for visual inspection, exploration and interpretation of data. We demonstrate the generalisability of PHLEX's modules on six types of healthy and tumour tissue, multiple multiplexed imaging technologies, formalin-fixed paraffin embedded (FFPE) and fresh frozen tissue sections, tissue microarray (TMA) and whole-slide images, and various antibody panels. As manual assessment, programming skills or pathology expertise are not essential, PHLEX addresses an unmet need for an end-to-end, user-friendly and scalable solution for IMC analyses. Its modules for phenotyping and spatial analyses can be applied to other multiplexed imaging technologies as standalone tools.

PHLEX is available at github.com/FrancisCrickInstitute/TRACERx-PHLEX with detailed documentation on tracerx-phlex.readthedocs.io.

## Results

### Overview of TRACERx-PHLEX
Here, we describe PHLEX, an automated workflow for cell PHenotype and Localisation analysis of multiplEXed imaging (Fig. 1a). Addressing key challenges with IMC data analysis, we developed and integrated three modules in the PHLEX workflow: (i) deep-imcyto for nuclear and whole-cell segmentation, (ii) TYPEx for in-depth identification of cell subtypes and cell states, and (iii) Spatial-PHLEX for interpretable spatial analysis. The modules are interoperable and encapsulated as independent pipelines in Nextflow, a workflow management platform with containerisation of all software dependencies[9]. Therefore, PHLEX's modules can also be used as standalone tools. The modules TYPEx and Spatial-PHLEX can be applied more broadly to multiplexed imaging data beyond IMC. We demonstrated the application of PHLEX in the TRAcking Cancer Evolution through therapy (Rx) (TRACERx) 100 cohort (ClinicalTrials.gov identifier: NCT01888601)[10] using IMC on 236 TMA cores from 83 patients with non-small cell lung cancer (NSCLC) (antibody panels, Supplementary Tables 1, 2)[11]. In addition, we evaluated and benchmarked its performance against standard tools using publicly available human and mouse data, IMC and Co-detection by indexing (CODEX) platforms, smaller TMA and larger whole-slide

images, and different tissue types, tissue fixation conditions and antibody panels (Fig. 1b). We analysed imaging data from tumour and tumour-adjacent lung, colorectal cancer, breast cancer, healthy intestine, Barrett's oesophagus and lymphoid tissue.

### deep-imcyto: nuclear and whole-cell segmentation for IMC
We previously described the development of the nf-core/imcyto IMC segmentation pipeline[12]. Here, we build on this work and present deep-imcyto (Fig. 2a), a flexible, automated deep learning-based Nextflow pipeline for cell segmentation of IMC images with multiple customisable workflow options. These options include data quality control and two cell segmentation workflows, with the choice dependent on the user's application needs. The module deep-imcyto addresses the need to automatically and accurately define cell boundaries and measure cellular marker intensities using IMC in advance of cell phenotyping and spatial analyses.

deep-imcyto uses highly accurate nucleus segmentation as the basis for whole-cell segmentation in IMC images, which it achieves through an instance-level nuclear segmentation process using an L4 UNet++ model architecture[13], trained on an in-house ground truth dataset, coupled with deep autoencoder-based anomaly detection (Supplementary Fig. 1a, b, Supplementary Fig. 2a–c). To address the need for annotated IMC datasets for model training, two experts manually labelled 41,962 morphologically diverse nuclei from TRACERx 100 IMC tissue cores, encompassing diverse lung cancer histology subtypes and architectures, as well as histologically normal lung, lymph node, tonsil and kidney tissues. We provide this TRACERx nuclear IMC segmentation dataset, to our knowledge currently the largest for IMC, as a resource for the research community (Data Availability, Supplementary Fig. 1b, Supplementary Methods).

Following deep-imcyto's nuclear segmentation procedure, we provide two workflow options for users to determine whole-cell boundaries based on the UNet++ nuclear seeds: *simple* and *CellProfiler* mode (Fig. 2a). For a fully automated cell segmentation procedure, deep-imcyto's default *simple* segmentation workflow dilates the deep learning model's nuclear masks by a user-defined number of pixels in order to approximate the cell boundary. This end-to-end option produces single-cell masks from input images without additional configuration. The *simple* workflow processes large input datasets rapidly: we ran 746 regions of interest (ROIs) from Jackson et al.[14]. with a mean execution time per ROI of 2.0 min. However, pixel dilation methods may fail to capture diverse cell morphologies or to perform well in areas of cell crowding, and cannot capture cells that lack in-plane nuclei. For these reasons, deep-imcyto, like nf-core/imcyto, allows a CellProfiler[15] pipeline to be supplied, performing user-defined cell segmentation steps after the identification of nuclei.

We used deep-imcyto's *CellProfiler* mode to perform a customised segmentation that uses the multiplexed channel information of IMC to improve identification of elongated stromal cells and aims to better discriminate the morphologies of the varied cell types of the lung tumour microenvironment—a schema we termed multiplexed consensus cell segmentation (MCCS, Methods, Supplementary Methods). We applied this approach to all TRACERx 100 IMC data. An example MCCS CellProfiler procedure is distributed with deep-imcyto and may be run on the PHLEX test dataset.

Resultant whole-cell masks from *simple* or *CellProfiler* modes are used to extract single cell-level marker intensity, cell-cell neighbouring data and morphometric features. As an additional feature, deep-imcyto generates pseudo-haematoyxlin-and-eosin (H&E) images from input IMC files without the need for ruthenium counterstaining[16] to enable direct pathologist labelling of IMC images (Fig. 2b). The single cell intensity data can be input into the cell phenotyping module, TYPEx, in the format of a cell-by-marker matrix for identification of cellular phenotypes and cell marker positivity status (Fig. 2c).

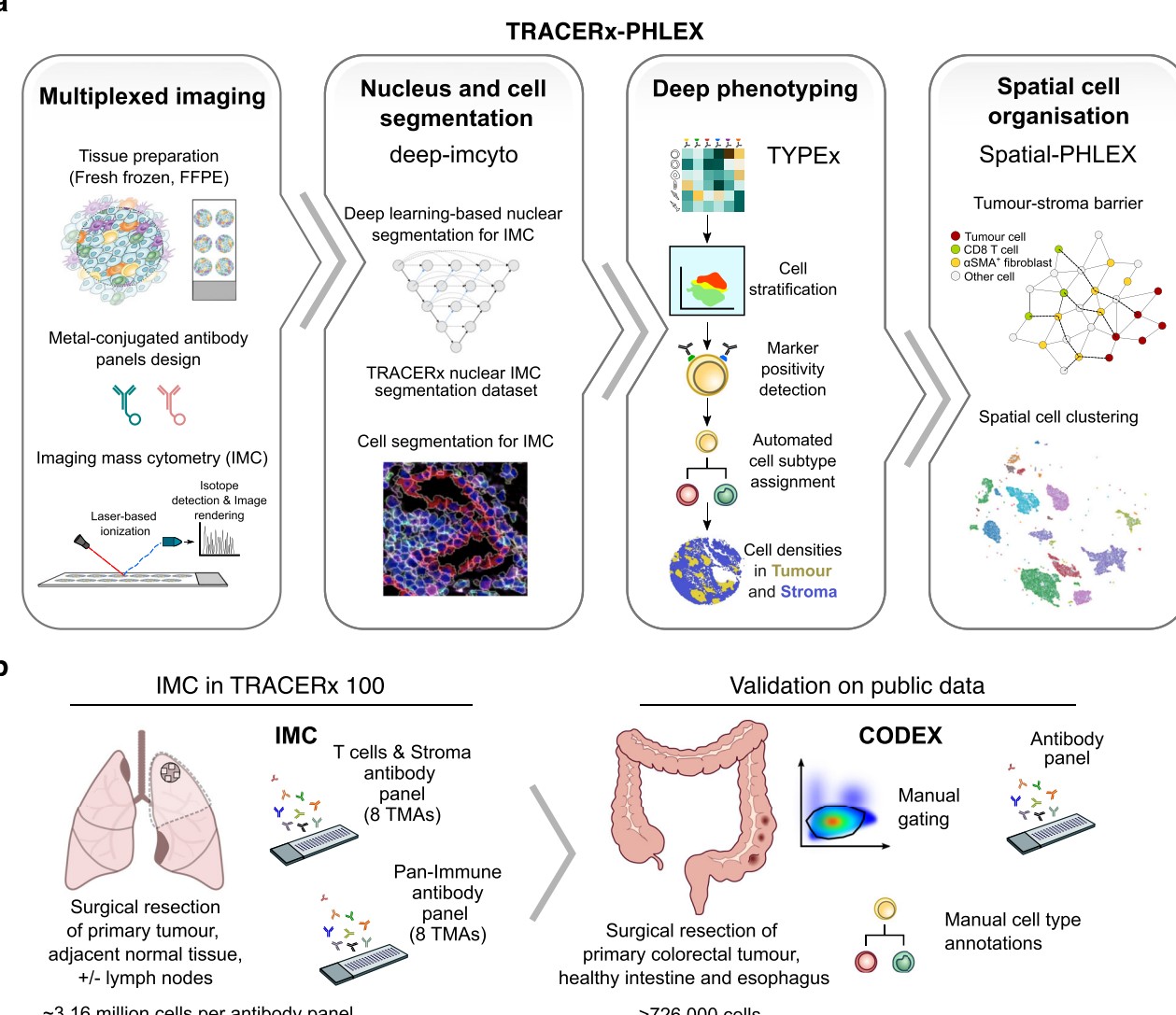

**Fig. 1 | TRACERx-PHLEX workflow overview and application in multiplexed imaging studies. a** PHLEX integrates three modules that cover the primary tasks in multiplexed imaging analysis: nucleus and cell segmentation (deep-imcyto), deep phenotyping (TYPEx) and spatial cell organisation analysis (Spatial-PHLEX). The module deep-imcyto performs image preprocessing, segmentation and image quality control. TYPEx annotates cell types and states on the basis of marker intensities. Spatial-PHLEX detects and quantifies spatial patterns in tissue organisation. **b** PHLEX was developed and applied on imaging mass cytometry (IMC) using resected tumour, lymph node and tumour-adjacent normal tissue from the TRACERx 100 cohort of patients with non-small cell lung cancer (NSCLC, $n = 60$

markers, 83 patients, 236 cores, ~3.16 million cells per antibody panel). Two antibody panels were profiled, a T cells & Stroma panel and a Pan-Immune panel. We validated PHLEX using orthogonal data within the TRACERx study and three public co-detection by indexing (CODEX) imaging datasets, which included manually curated cell type annotations. The colorectal cancer dataset also included manual gating information (Schürch et al., $n = 56$ markers, 140 TMA cores from 35 CRC patients). The datasets from Barrett's oesophagus (Brbić et al.) and healthy intestine (HuBMAP) included fresh frozen, whole-slide tissue sections ($n = 44$–48 markers, >726,000 cells). FFPE formalin-fixed paraffin-embedded, TMA tissue microarray.

## In-depth identification of cell phenotypes in the tissue microenvironment with TYPEx

The cell phenotyping module, TYPEx, addresses the need to automatically identify and assign cell types and comprehensively detect cell states of individual, segmented cells. To do this, TYPEx detects positivity of markers, resolves low-confidence annotations and assigns a cell subtype based on a combination of positive markers on a cell (Fig. 3a). The required input is a matrix of cell-by-marker intensities and a file with cell-type definitions tailored to the user's antibody panel, a format that allows flexibility to other imaging modalities. TYPEx first builds on existing tools to perform cell stratification and then implements a statistical approach to determine marker positivity. Based on the combination of positive markers, it automatically annotates cell subtypes according to the user-provided cell type definitions.

Here, we elaborate on the four analytical steps of TYPEx: cell stratification, marker positivity detection, cell subtype assignment, and tissue segmentation, as demonstrated with the TRACERx 100 IMC dataset.

The cell-stratification step aims to divide the cells into groups with similar marker intensity levels and differentiate between low- and high-confidence groups based on the intensities of major cell-lineage markers. Low-confidence cells can localise in densely packed immune areas, such as lymphoid aggregates, where the signal can spill over into neighbouring cells (Supplementary Fig. 3). In addition, low-confidence cells can have weak intensities and low signal-to-noise ratio of cell lineage-specific markers (Supplementary Fig. 4). The three-tiered cell-stratification approach in TYPEx combines existing probabilistic and unsupervised methods to stratify first by putative major cell lineage, second by confidence, and third by cell clustering (Fig. 3a).

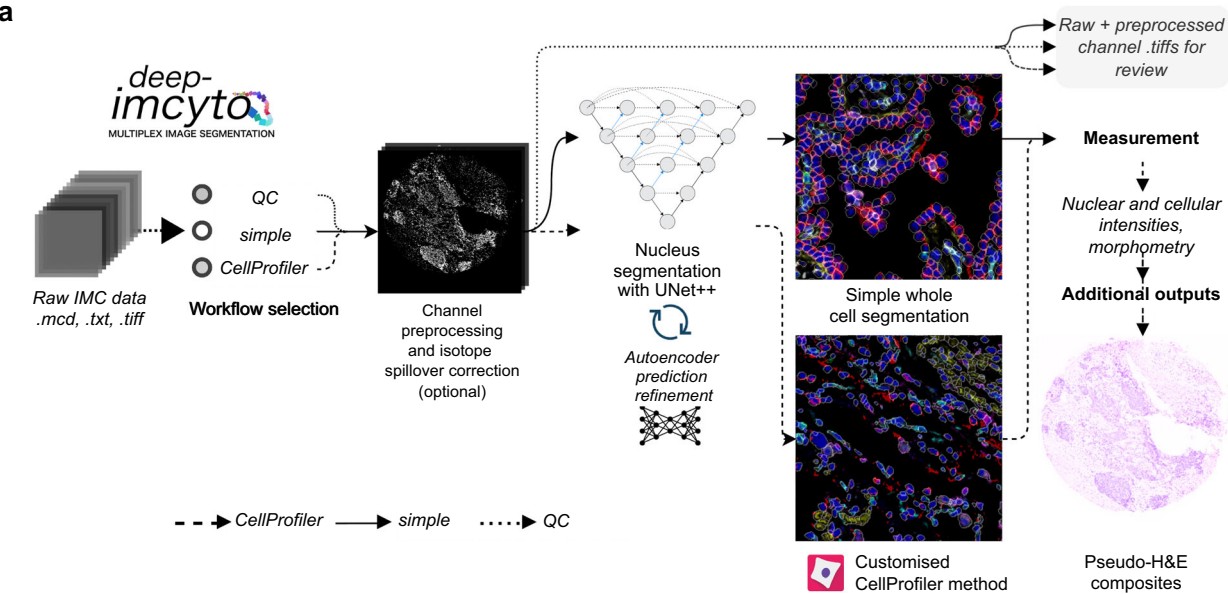

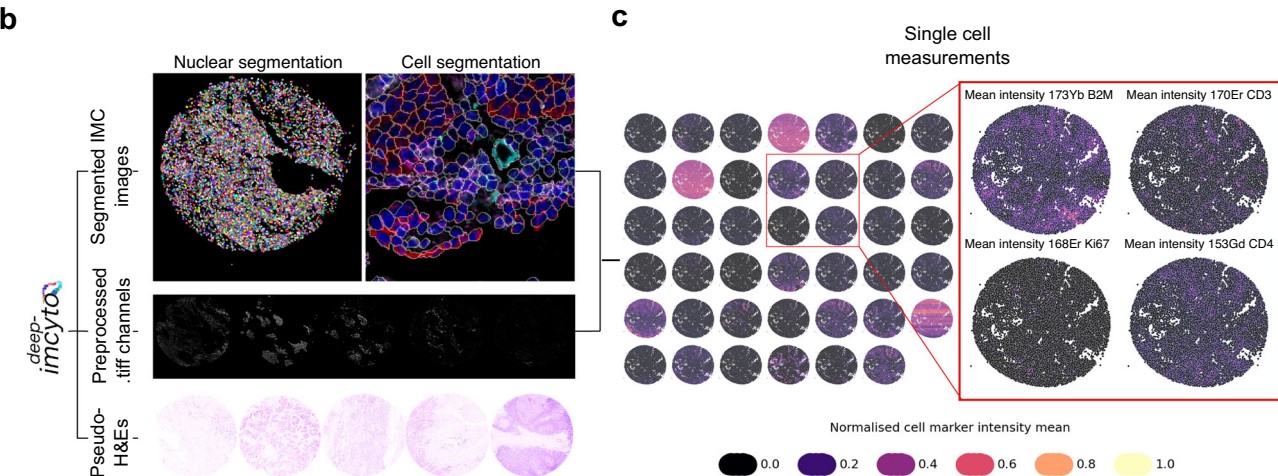

**Fig. 2 | The deep-imcyto segmentation pipeline for IMC images. a** Overview of the processing workflows available in deep-imcyto: *QC*, *simple* segmentation and *CellProfiler* segmentation. **b** Example standard image outputs of the deep-imcyto segmentation workflows: [1] nuclear and whole-cell segmentation masks for input IMC images, [2] preprocessed channel images, [3] pseudo-H&E images constructed from intensity projected IMC channel data. The whole-cell segmentation mask was generated using deep-imcyto in *simple* mode. The cell outlines were overlaid on an IMC composite image (red pancytokeratin, green CD8a, cyan CD45, blue DNA, magenta CD4, yellow CD31/αSMA). **c** Single cell spatial plots produced by the deep-imcyto *simple* workflow give the user a quick look at the spatial distribution of all markers in their experiment. All markers are normalised independently per image for visualisation purposes. H&E haematoxylin and eosin.

First, to stratify by major cell lineage, TYPEx creates a probabilistic model using CellAssign[17], a hierarchical statistical framework that computes the probability of a cell being of a given cell type using an expectation-maximisation algorithm. TYPEx further stratifies cells by the confidence in major cell lineage assignment (Fig. 3a), relying on the assumption that robust cell annotations are more likely to retain their label and low-confidence annotations to change it when perturbing the input to the model. By providing partial information to the probabilistic model, TYPEx identifies examples of stable and variable cell annotations between the different models for each cell lineage (Methods). A binomial logistic regression model is fit to separate stable from variable major cell lineage calls based on their probability score and the lineage-specific marker intensities. This model is then used to predict low- and high-confidence cells. Overall, in the TRACERx 100 IMC dataset, low-confidence cell type calls accounted for 28.5% and 29.7% of cells in the two IMC panels, although at variable rates among images.

Finally, the last cell-stratification step applies a standard clustering approach, PhenoGraph[18] or its variant FastPG[19], within each group of cells previously stratified by major cell lineage and confidence level. The clustering is performed on all marker intensities within each stratified group, which allows exhaustive detection of protein expression, and identification and discovery of cell subpopulations.

To avoid setting a global cutoff on signal intensities for detecting marker positivity, the marker intensities on a given cluster are transformed into probabilities - the probability that a random cell from the cluster has a higher intensity of a given marker than a random cell from another cluster. The deviation of the probability distribution of the given cluster from a background distribution is summarised with a D-score, which represents the difference in area between the cumulative probability distributions (Fig. 3b, Methods). A D-score is assigned to each cluster for each measured marker. A cluster that expresses a given marker will have a high D-score reflecting the skewed probability distribution towards higher values compared to the background distribution.

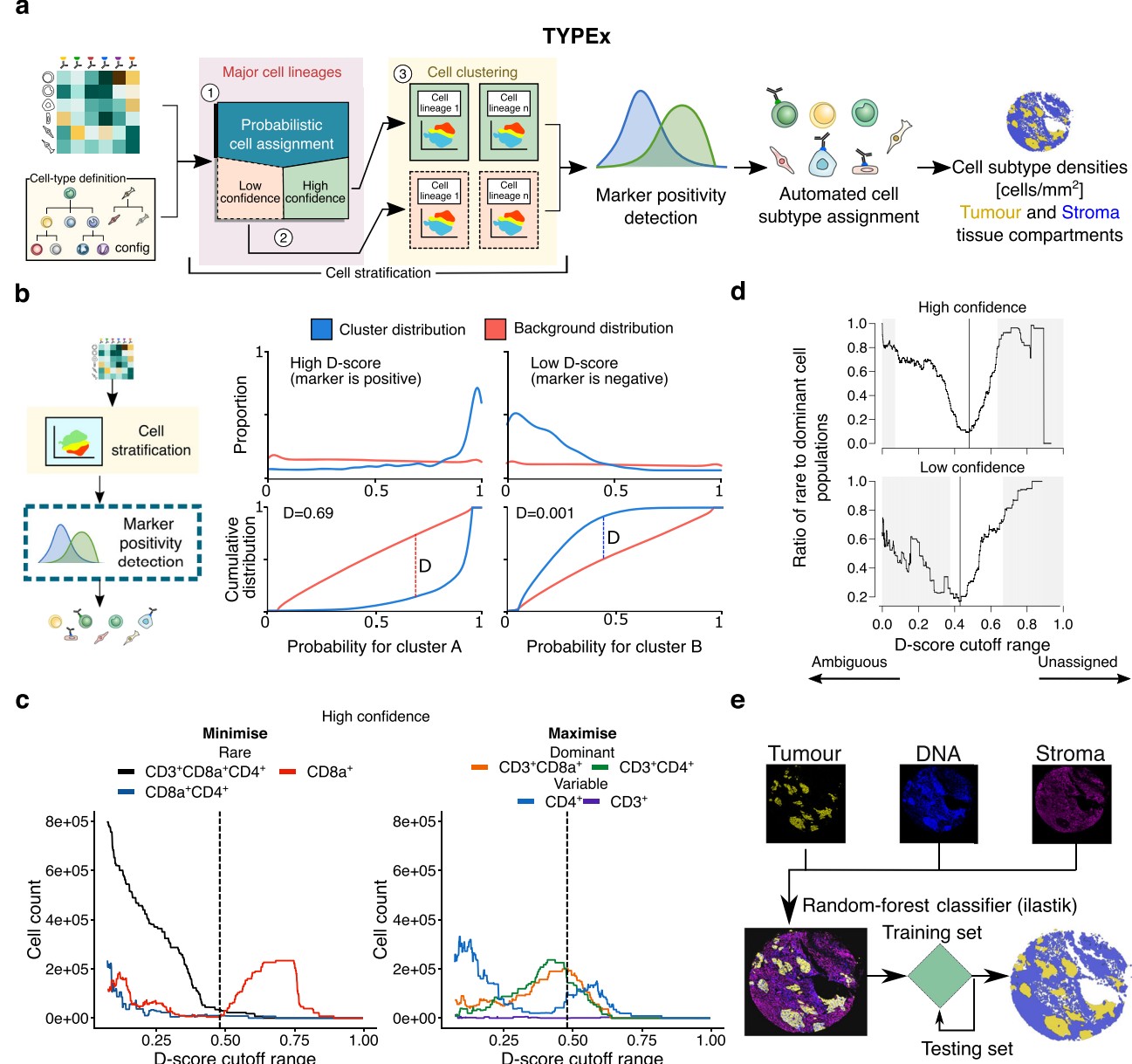

**Fig. 3 | Automated deep cell phenotyping from multiplexed imaging using TYPEx. a** Using a cell-by-marker intensity matrix and cell-type defining config file as input, TYPEx performs four steps: cell stratification through combination of existing methods (1–3), marker positivity detection, cell-type assignment and tissue segmentation. **b** To determine marker positivity, each cluster derived from the cell-stratification step is compared pairwise with all other clusters, and for each marker, the probability that a random cell from the given cluster has a higher intensity of that marker than cells from another cluster is calculated. Examples of probability distributions for a cluster A expressing a given marker (top left) and a cluster B that does not express that marker (top right) are illustrated. The D-score represents the maximum positive distance from the cumulative to the background distribution (bottom). **c** For each confidence group and across all possible D-score cutoffs (0–1 range, step 0.0001), the number of cells expressing a combination of user-provided markers is calculated. **c** illustrates an example for the high-confidence group in the

T cells & Stroma panel using the default T-cell markers, based on which, three types of T-cell populations are defined: rare, dominant, and variable (vary depending on the dataset). The optimal cutoff minimises the rare (left) and maximises the dominant (right) T-cell subpopulations. **d** The ratio of rare to dominant T-cell populations against the range of D-score cutoffs. The cutoff range in which any of the dominant populations has zero cell count is not considered (grey area). At the lowest values of the D-score cutoff, the number of double-positive (CD3$^{+/-}$) CD8a$^+$CD4$^+$ T cells and overall Ambiguous cells increases; as the D-score cutoff exceeds the optimal, the number of single-positive CD3$^-$CD8a$^+$ T (overall Unassigned cells) increases. The optimal D-score cutoff, shown with a vertical black line (**c**, **d**), is determined individually for the low- and high-confidence groups in each study and panel. **e** To output cell densities, TYPEx uses a random forest classifier for tissue segmentation, a user-specific model, or binary masks of tissue domains as input. Source data are provided as a Source Data file.

TYPEx calculates an optimal D-score cutoff and calls positivity of a marker on a given cluster if the cluster's D-score for this marker is higher than the cutoff. The optimal cutoff is determined based on known co-expression patterns of a subset of markers. By default, TYPEx uses the T cells markers CD4, CD8a and CD3, where double-positive CD4$^+$CD8a$^+$ (CD3$^{+/-}$) and single-positive CD8a$^+$ cells, for

example, are expected to be rarely found in peripheral non-lymphoid tissue, whereas CD3$^+$CD4$^+$CD8a$^-$ and CD3$^+$CD4$^-$CD8a$^+$ cells are expected to be the dominant populations in the analysed cohort (Fig. 3c). Based on these criteria, TYPEx estimates an optimal D-score threshold as the minimum value of the ratio of rare to dominant T-cell populations, minimising the rare and maximising the dominant T-cell

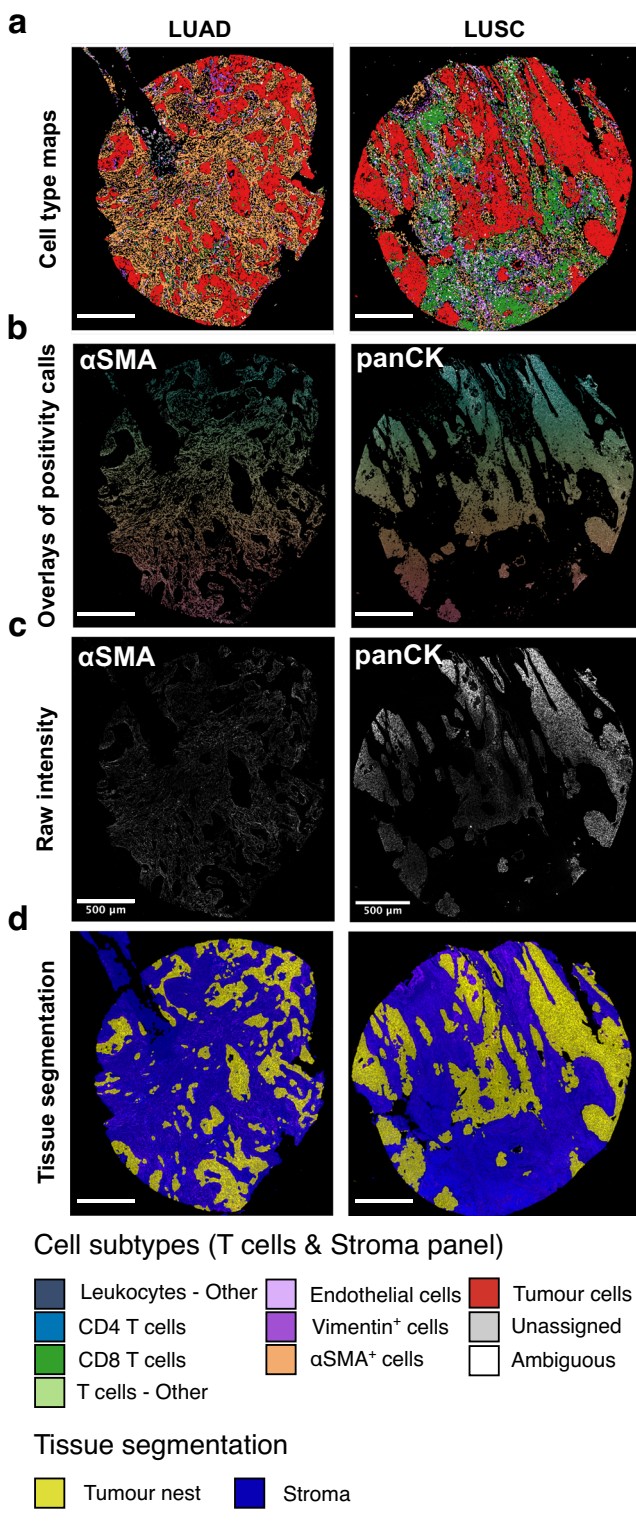

**a**

LUAD    LUSC

Cell type maps

**b**

Overlays of positivity calls

αSMA    panCK

**c**

Raw intensity

αSMA    panCK

500 µm    500 µm

**d**

Tissue segmentation

## Cell subtypes (T cells & Stroma panel)

- Leukocytes - Other
- CD4 T cells
- CD8 T cells
- T cells - Other
- Endothelial cells
- Vimentin⁺ cells
- αSMA⁺ cells
- Tumour cells
- Unassigned
- Ambiguous

## Tissue segmentation

- Tumour nest
- Stroma

**Fig. 4 | Example image outputs from TYPEx in the TRACERx 100 IMC dataset.** TYPEx outputs various images for the user, examples of which are shown for two cases from the PHLEX test dataset (T cells & Stroma panel), including: **a** A map of cell objects coloured by cell subtype for each analysed sample. **b**, **c** For each major cell lineage marker, the samples with the highest cell counts are selected for visual inspection. The positive cell objects for a corresponding marker are overlaid onto a raw single-channel intensity image (**b**). Each cell object is visualised with a different colour. A raw single-channel intensity image for the markers in major cell lineage definitions is also provided (**c**). **d** Tissue segmentation masks based on tumour- and stroma-specific markers. The cell maps and overlays in **a**, **b** are generated when the mask of segmented cell objects is provided as input for TYPEx. Scatter plots of annotated cell objects are output as an alternative. LUAD lung adenocarcinoma, LUSC lung squamous cell carcinoma.

Cell annotation involves the estimation of a cell subtype specificity score based on the combination of expressed markers and according to user-provided cell subtype definitions. The cell subtype with the highest specificity score is assigned (Supplementary Fig. 5, Supplementary Methods). When a cell is Unassigned or Ambiguous, TYPEx considers the high-confidence major cell lineage assignments from CellAssign and the tissue segmentation when available.

To quantify the cell annotations per tissue area and within tumour and stroma tissue compartments, we trained a three-class random forest model for background, tumour nest/epithelium and stroma region segmentation on images from tumour, adjacent-normal lung and lymph node tissue using ilastik[20] (Fig. 3e, Methods). To allow broader use across different tissue types, TYPEx also incorporates an alternative two-class model for tissue and background segmentation. Additionally, it accepts as input user-provided binary masks, such as pathology annotations, and labels the cell objects within these regions (Supplementary Fig. 6). The time and computational complexity for cell phenotyping can vary depending on the number of cells, the number of measured markers and the number of major cell lineages in the dataset (Supplementary Table 3).

TYPEx outputs various summary tables, graphs, visualisations of pixel-level information and maps of annotated single-cell objects to facilitate data interrogation, interpretation and quality control of the cell phenotyping results (Fig. 4). The output cell objects tables can be used as direct input for Spatial-PHLEX.

### Identification of spatial features with Spatial-PHLEX

Spatial-PHLEX is a toolkit of spatial analysis methods for cell coordinate data. Provided as a self-contained Nextflow pipeline that takes single cell location and phenotype data in a predefined, simple format, Spatial-PHLEX has two core subworkflows: spatial clustering and graph analysis of tissue barrier structures (Fig. 5a–c).

Spatial dependency between cells occurs over a range of distance scales. Several tools exist for probing a cell's immediate neighbourhood[21,22], but important biological coordination can also be present at longer distance scales[23,24]. To examine the role of cellular niches, spatial clustering has been introduced to digital pathology and multiplexed imaging in recent years[3,25,26]. For example, it has been used to examine chemokine expression in B cell niches in metastatic melanoma[25], while Varrone et al. have recently applied the technique to spatial transcriptomics data[26]. Spatial-PHLEX implements density-based spatial clustering by applying the DBSCAN algorithm[27] to phenotyped cell coordinate data on a cell type-specific basis, generating, for example, tumour cell and CD8 T-cell spatial clusters (Fig. 5b). Spatial-PHLEX then calculates the boundary of each spatial cluster, and the composition of other cell types located within it. This approach allows the discovery of spatial niches of a chosen cell type and provides a detailed breakdown of the other cell types contained within these niches. Such a measure informs the extent to which a given cell type may be internalised within a dense cluster of another cell type, such as CD8 T cells within a tumour cell cluster. Standard output of

populations. Overestimating the D-score cutoff results in a higher proportion of single-positive CD8a⁺ cells and overall Unassigned calls, whereas underestimating the D-score cutoff results in a higher proportion of double-positive CD4⁺CD8a⁺ T cells (CD3⁺/) and overall Ambiguous calls (Fig. 3d). Notably, the D-score threshold selection is unbiased towards other rare or frequent populations in the cohort. Optimal thresholds are determined individually for the low- and high-confidence cells and applied across all clusters for each confidence group in the dataset. These thresholds are estimated individually for each dataset, antibody panel and cohort without manual intervention.

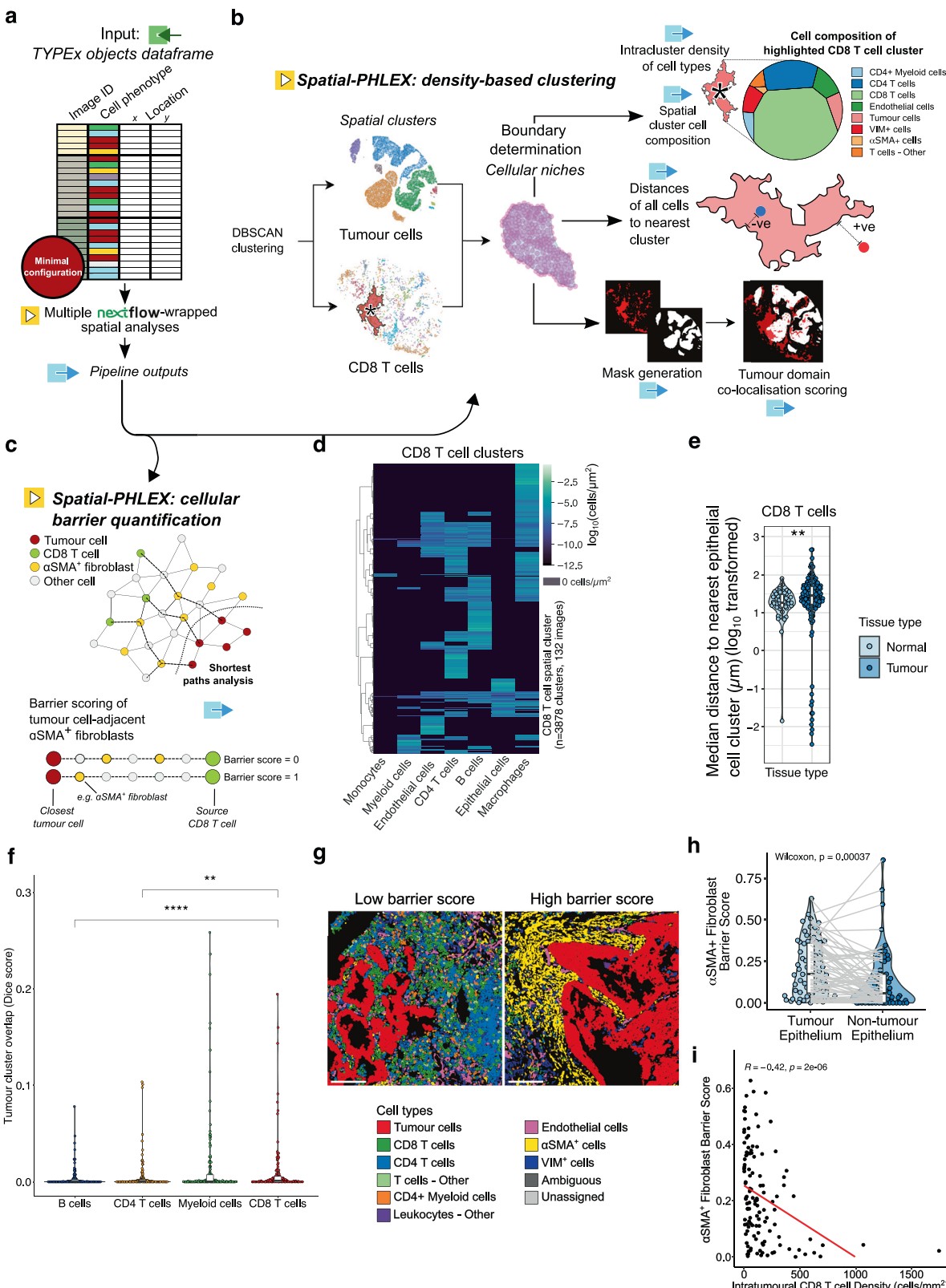

Spatial-PHLEX includes single-cell tables of cluster membership and the intracluster density of all cell types, plus the distances of all cells to the edge of the nearest cluster. Furthermore, Spatial-PHLEX outputs binary masks of all identified clusters, from which cluster co-localisation metrics are determined as a measure of interaction between dense niches of cell types. The spatial clustering workflow typically executes in under a minute for a single input cell type.

Spatial-PHLEX is also able to interrogate the cellular network present in a dataset in order to measure the degree to which a 'barrier' cell type may be impeding one cell type from direct access to a second cell type (Fig. 5c, Supplementary Fig. 7). Failmezger et al.[8] introduced this approach and the "stromal barrier" metric to quantify how stromal cells impede the access of lymphocytes to tumour nests in metastatic melanoma and demonstrated that high barrier scores are associated

**Fig. 5 | The Spatial-PHLEX analysis pipeline. a** Spatial-PHLEX runs multiple spatial analyses with simple input and minimal configuration. **b** Its density-based clustering workflow applies the DBSCAN and alpha shape algorithms to cell position data to find dense domains of a given cell type. Outputs include (i) intracluster densities of cell types, (ii) spatial cluster composition, (iii) distances of all cells to the boundary of the nearest cluster, with a negative distance assigned to cells within the spatial cluster and a positive distance to those outside, (iv) masks for clusters, and (v) overlap metrics for spatial clusters of different cell types. **c** Conceptual overview of Spatial-PHLEX barrier scoring and the concept of barrier fibroblasts between CD8 T cells and tumour. bottom, Illustrative barrier quantification from a shortest paths analysis. **d** Intracluster densities of cell types in CD8 T-cell spatial clusters in the TRACERx 100 cohort (Pan-Immune panel; $n = 3878$ clusters from $n = 130$ tumour cores, $n = 2$ benign tumour-adjacent cores). **e** $Log_{10}$-transformed image-level median distance-to-nearest-epithelial-cell-cluster measurements for CD8 T cells in tumour versus normal tissue ($n = 139$ tumour, $n = 46$ normal cores). Two-tailed Wilcoxon $p = 0.0045$. **f** Quantification of immune cell cluster and tumour cell cluster co-localisation through Dice scoring ($n = 136$ tumour cores). Two-tailed Wilcoxon $p$ values: CD4 T cells $p = 0.006$, B cells $p = 1e-4$. **g** Example low and high $\alpha$SMA$^+$ fibroblast barrier score cases. Scale bar = 150 $\mu$m. **h** Paired violin plots of barrier scores for CD8 T cells to non-tumour and tumour epithelial cells within the same tumour cores ($n = 67$ cores). Paired two-tailed Wilcoxon $p$ value shown. **i** Spearman correlation of barrier score vs CD8 T-cell density in the lung tumour tissue compartment ($n = 121$ cores). All boxplots show median and lower and upper quartile values, and whiskers extend up to 1.5*IQR above and below quartiles. **h, i** The all-paths adjacent barrier fraction score and tumour cores from LUAD and LUSC histologies are used. $P$ values are represented by: *$p < 0.05$, **$p < 0.01$, ***$p < 0.001$, ****$p < 0.0001$. No adjustments for multiple testing. Source data are provided as a Source Data file.

with poorer prognosis. In Spatial-PHLEX, we implement a version of the stromal barrier score that capitalises on the extensive phenotypic detail available from multiplexed imaging to generate barrier scoring for any triplet of cell types available in the input dataframe. The overall barrier score for an image is the average barrier score calculated over all cells of the specified source cell type, the distribution of which may be measured across an experimental cohort (Supplementary Fig. 7a). Spatial-PHLEX computes multiple alternative barrier score metrics for a given sample (Supplementary Fig. 7b). Spatial-PHLEX also produces a summary table for a given image with barrier and path information for all the cells for which the barrier is measured. The typical time it takes to perform the barrier scoring workflow is 10 min and scales linearly with the number of input cells.

### Spatial-PHLEX quantifies barrier-like features to CD8 T-cell infiltration in non-small cell lung cancer

In our work, we used Spatial-PHLEX to derive spatial clusters of both tumour cells and CD8 T cells across TRACERx 100 tumour cores (Pan-Immune IMC panel) to illustrate the spatial insights gained from our approach related to CD8 T-cell infiltration, which is prognostic in several cancer types[28,29]. Using intracluster density outputs, macrophages were the other cell type most frequently observed within CD8 T-cell clusters (55% of CD8 T-cell clusters containing ≥1 macrophage), although phenotypically diverse clusters were also frequently observed (Fig. 5d). Furthermore, the distance between CD8 T cells and epithelial cell clusters was significantly higher in tumour cores compared to in normal tissue cores (two-tailed Wilcoxon $p = 0.0045$, Fig. 5e). We observed that CD8 T-cell clusters overlapped with tumour cell clusters to a greater extent than other immune cell types did based on the calculated Dice score for overlap of the corresponding spatial cluster masks (two-tailed Wilcoxon tests, unadjusted: CD4 T cells $p = 0.006$, B cells $p = 1e-4$; Fig. 5f).

We performed Spatial-PHLEX barrier scoring for the same cohort and calculated the barrier score based on the interpositioning of $\alpha$SMA$^+$ fibroblasts between CD8 T cells and tumour/non-tumour epithelial cells (TRACERx 100, T cells and Stroma panel). Generally, low and high scores reflected what would be expected from the prevalence and spatial localisation of the barrier cell type by eye (Fig. 5g), and this also agreed with simulated data of a simplified barrier model (Supplementary Fig. 7c–e). A paired analysis of 67 TMA cores with presence of both non-tumour and tumour epithelial cells, as defined by pathology review, revealed a significantly higher barrier around tumour epithelial cell clusters than non-tumour epithelial cell clusters (two-tailed Wilcoxon paired test, $p = 0.0004$, Fig. 5h). Furthermore, the density of CD8 T cells within the tumour nest tissue compartment was negatively correlated with the $\alpha$SMA$^+$ fibroblast barrier score indicating that the barrier score may reflect physical exclusion of CD8 T cells by $\alpha$SMA$^+$ fibroblasts (Spearman $\rho = -0.42$, $p = 2e-06$, Fig. 5i).

### deep-imcyto is a reliable segmentation model for IMC data

We benchmarked PHLEX's segmentation and phenotyping modules against relevant alternative approaches in the field. We first benchmarked deep-imcyto's nuclear segmentation model over 19 segmentation quality metrics against several publicly available models that have been applied to IMC data. These included the generalist cell segmentation models Mesmer[30] and Cellpose[31], and two nuclear segmentation models distributed with Stardist[32], the DataScienceBowl 2018 model and the versatile model. We also compared deep-imcyto against an IMC-specific Stardist model, which we trained on the same data as deep-imcyto. Overall, deep-imcyto performed well compared to these approaches (Fig. 6a, b, Supplementary Fig. 8). It significantly outperformed the nuclear predictions of generalist cell segmentation model Mesmer in 10 of 19 metrics, 11 of 19 metrics for Nuclear Cellpose, and significantly outperformed all out-of-the-box models in nine of the 19 metrics. deep-imcyto outperformed Stardist when using either of the distributed models; however, performance between the models was similar for 11 metrics once Stardist was retrained for IMC applications using the TRACERx ground truth data. Based on visual assessment of IMC images, deep-imcyto provides good quality nuclear segmentations comparable to these commonly used deep learning segmentation approaches (Supplementary Fig. 9a). We applied deep-imcyto's nuclear model to three publicly available IMC datasets representing different tissue types (breast and mouse lung cancer) and different fixation conditions (fresh frozen, FFPE)[12,14,30]. Despite that the deep-imcyto training dataset mostly comprises FFPE human lung tissue, deep-imcyto performs qualitatively well on the different public datasets (Supplementary Fig. 9b). Taken together, deep-imcyto is a ready-to-go solution for segmenting IMC images to high accuracy and, when analysing IMC data, IMC-specific nuclear segmentation models should be preferred over generalist models trained on TissueNet[30], which has low representation of IMC annotations compared to those from other imaging modalities.

Assessment of whole-cell segmentation was performed qualitatively. We compared deep-imcyto's *simple* and the MCCS procedure within the *CellProfiler* workflow for TRACERx 100 lung cancer tissue to that achieved using Mesmer (Fig. 6c). Mesmer and *simple* segmentation produced highly visually similar results, with both techniques capturing comparable cellular features. We found MCCS to improve the capture of cell body features over Mesmer in cases of irregular morphology, such as highly elongated endothelial cells, and to better capture the extent of non-nucleated stromal content derived from the $\alpha$SMA channel (putative fibroblasts) (example in Fig. 6c, third column). The *simple* workflow is an effective form of cell segmentation for users who wish to run deep-imcyto without further configuration, or those whose samples lack cells with the more complex properties that may benefit from a bespoke *CellProfiler* approach.

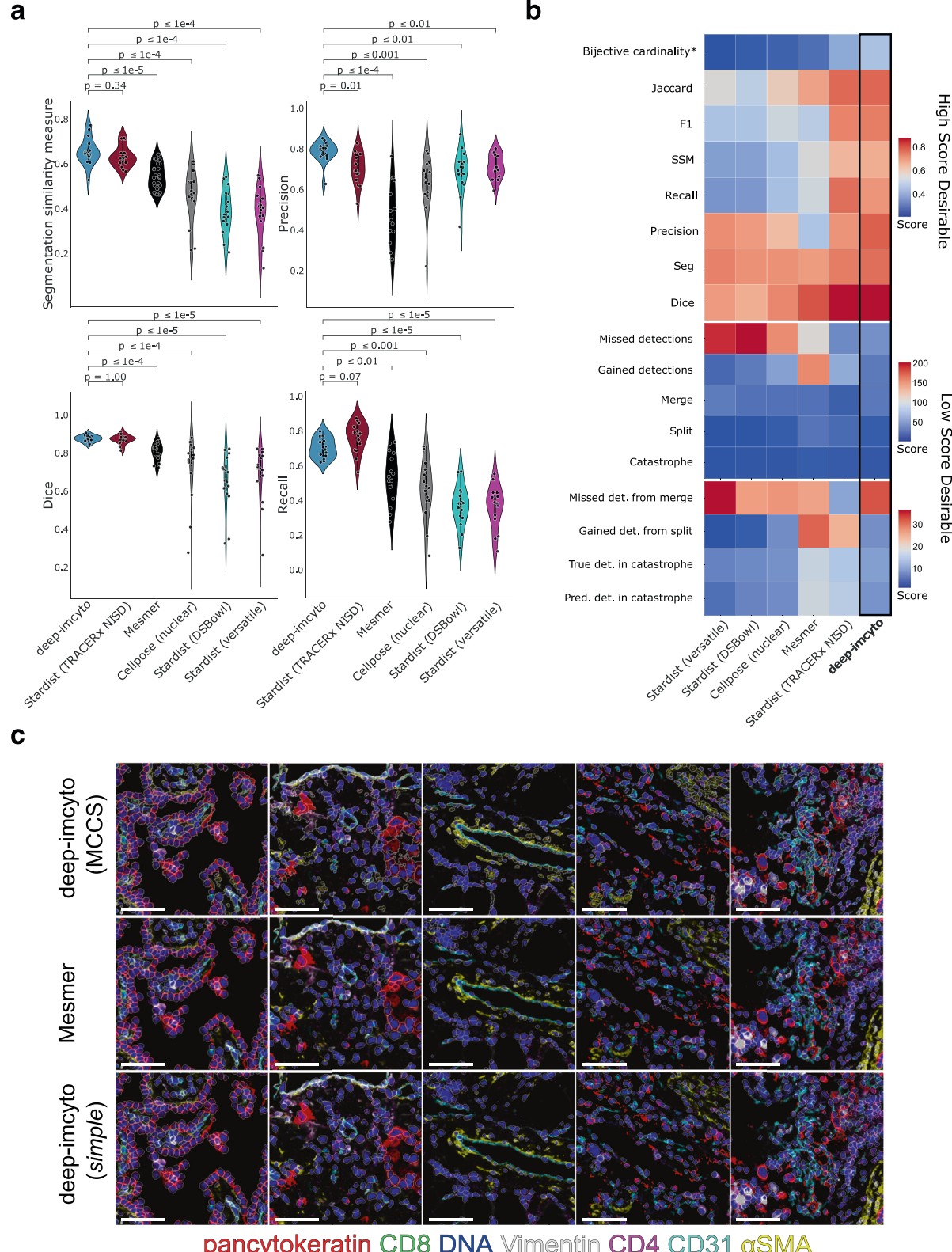

**Validation and benchmarking of cell phenotyping with TYPEx**
The performance of TYPEx was evaluated using matched orthogonal data in the TRACERx 100 IMC cohort and manually curated annotations of publicly available CODEX imaging data from colorectal cancer[22], Barrett's oesophagus[7] and healthy intestine[33].

For orthogonal validation, we used whole-exome sequencing (WES), immunohistochemistry (IHC), histopathology scoring and flow cytometry from matched tumour samples in the TRACERx 100 cohort

(Fig. 7a–c, Supplementary Fig. 10a–c). We observed a significant positive correlation of TYPEx with the WES-derived T-cell estimates using the T cell ExTRECT tool[34] (Spearman correlation, $\rho = 0.49$), pathologist-labelled tumour-infiltrating lymphocyte (TIL) scores[35] from paired H&E images ($\rho = 0.57$), and CD3 IHC staining in both the stroma tissue compartment ($\rho = 0.7$) and pathologist-annotated area of malignant tumour ($\rho = 0.48$). At a more granular level, the CD8a$^+$ T-cell subpopulations defined by five markers, CD103, CD57, CD27,

**Fig. 6 | Evaluation of PHLEX segmentation performance compared to standard approaches. a** Instance segmentation similarity metric (Al-kofahi et al.), Dice, precision and recall scoring performance of deep-imcyto nuclear segmentation *vs* other publicly available methods: Mesmer, Cellpose, the Stardist "versatile" model and Stardist trained on DSBowl 2018 data, as well as a Stardist model retrained with the TRACERx nuclear IMC segmentation dataset (TRACERx NISD). Each score is calculated per image, and the test dataset covers 6453 nuclei across *n* = 16 TRACERx NISD images, which were not included in the training of any of the models. Significance values indicate the results of a two-tailed Mann−Whitney *U* test.
**b** Heatmap summary of the mean segmentation performance of each metric shown in Supplementary Fig. 8. Upper panel shows scores, where higher values indicate superior performance. The lower two panels show scores, where a lower value is

indicative of a better performance. *Bijective cardinality was normalised by the total possible number of correct detections in the test dataset. **c** Qualitative comparison between the deep-imcyto *simple* segmentation workflow (1 pixel dilation) and Mesmer, as well as the MCCS procedure run in deep-imcyto's *CellProfiler* mode. All methods perform well at identifying cellular material; however, MCCS captures challenging cell morphologies and identifies non-nucleated stromal cell content (αSMA - putative fibroblasts in yellow and CD31 - endothelial cells in teal). Five example tiles from five different tissue cores (three tumour, one benign tumour-adjacent, one lymph node) from the TRACERx 100 study (T cells & Stroma antibody panel). All tiles are 256 × 256 μm, scale bar = 75 μm. Box plots in (**a**) show lower and upper quartile values, and whiskers extend up to 1.5*IQR above and below the quartiles. Source data are provided as a Source Data file.

CD45RA and CXCR6, had comparable proportions between flow cytometry[36] and IMC data of matched tumour regions (Supplementary Fig. 10b). The strongest correlation was observed for the largest T-cell subpopulation, CD8$^+$ tissue resident memory T cells ($\rho = 0.57$, Supplementary Fig. 10c).

To assess the reproducibility of TYPEx, we compared TRACERx IMC data from two distinct antibody panels on matched TMA cores. We observed a positive correlation of the shared phenotypic markers between the two panels, ranging from 0.52 for CD103$^+$ cells to 0.88 for CD8a$^+$ cells (Fig. 7d). Strong positive correlation between two panels was observed for the major cell lineages and cell subtype annotations (Supplementary Fig. 10d, e).

We also analysed independent public CODEX imaging datasets for TYPEx validation using different tissue types, fixation conditions, antibody panels, and image sizes. Using manually gated marker positivity on a colorectal cancer (CRC) CODEX imaging data of FFPE-embedded TMAs[22], we demonstrated a strong correlation for 67% of the markers analysed with manual gating (Spearman correlation coefficient $\rho > 0.8$) and a significant moderate correlation for the remaining 33% ($\rho > 0.5$) independent of the cell population size (Supplementary Fig. 11a). Furthermore, TYPEx performed comparably to published manually curated annotations derived from clustering with X-shift[37] on the CRC dataset, based on their correlation with cell annotation counts from manual gating (Fig. 7e). The median correlation coefficient was 0.73 for TYPEx and 0.63 for the curated X-shift annotations. We observed a concordance of 83% between TYPEx and curated X-shift annotations for the unambiguous cell assignments (Supplementary Fig. 11b).

To assess the performance of TYPEx on larger images from fresh frozen tissue sections, we used published ground truth annotations on Barrett's oesophagus (BE) and healthy intestine tissue (HuBMAP) derived from iterative clustering and manual curation[7]. The TYPEx annotations of the BE dataset matched 80% of the ground truth annotations, among which 2% were Ambiguous or Unassigned calls (Supplementary Fig. 12a, b). Nearly half of the discordant annotations were labelled stroma in the reference data and identified as CD34$^+$ Endothelial cells by TYPEx.

Using the more granular annotations on the HuBMAP dataset (patient B004), TYPEx identified all the cell subtypes and epithelial states from the expert-annotated cell subtype labels. Among the unambiguous cell subtype assignments, 85% of the cell subtypes and 75% of the cell states matched the ground truth (Supplementary Fig. 12c). 41% of the discordant assignments were epithelial cell states, which may be attributed in part to differences in the criteria for defining these cell states.

Finally, we benchmarked the performance of TYPEx against clustering, probabilistic, and deep learning approaches for cell phenotyping. When comparing the proportion of Ambiguous or Unassigned cells identified in the TRACERx 100 IMC cohort, FastPG identified 44% Ambiguous or Unassigned cells, compared to 4.7% with TYPEx (Fig. 7f, T cells & Stroma IMC panel). For example, we observed a large proportion of double-positive CD4$^+$CD8a$^+$ T cells with FastPG

compared to TYPEx (Fig. 7g). In the Pan-Immune panel, the percentage of Unassigned cells was significantly reduced to 19% with TYPEx from 27−30% with FastPG and PG. The proportion of Unassigned cells remained higher in this antibody panel compared to T cells & Stroma, because a fraction of non-immune stromal cells were not targeted by any marker in this panel (Supplementary Table 2). Using the CRC dataset, we demonstrated that the proportion of Ambiguous or Unassigned cells detected with TYPEx matched the published manually curated annotations (12.2% compared to 15.9%, respectively) and was twice lower compared to FastPG and PhenoGraph without manual adjustments and any stratification steps (25−26%, Fig. 7f). Furthermore, when applied to IMC data, FastPG and PhenoGraph consistently assigned 63−71.5% to the same major cell lineage between subsampling runs on the TRACERx 100 IMC dataset (2/3 random subsampling, $n$ = ~2.1/3.16 million cells, Supplementary Fig. 13a) and 58−69% in the CRC data ($n$ = 172,257/258,385 cells, Supplementary Fig. 13a, b), significantly lower than the concordance of 91−95% reached with the supervised approach, CellAssign. These results support manual splitting and merging of clusters when analysing multiplexed imaging data using clustering approaches[14,22]. Importantly, these analyses also demonstrated that the TYPEx stratification steps were crucial to maximise the fraction of assigned cells and reduce ambiguity.

Based on the evaluation metrics, macro F1-score and mean precision and recall, the performance of TYPEx was comparable to the probabilistic cell phenotyping approaches CELESTA[5] and Astir[6] and the geometric deep learning method STELLAR[7] (Fig. 7h−j, Supplementary Fig. 13c−e). The macro F1-score on the CRC dataset ranged from 0.5−0.6 for TYPEx, CELESTA and STELLAR, and 0.45 for Astir. In this dataset, CELESTA scores for TMA A were derived from a published confusion matrix. The TMA A scores were higher than those for TMA B, which were derived from our analysis using the same settings for TMA A. These results suggest that the optimisation of the cell phenotyping tools and the user's familiarity with them affects their performance, and CELESTA may require optimisation for each TMA. In the BE dataset, the macro F1-score, mean precision and mean recall for TYPEx (0.7−0.74) were comparable to the ones achieved by STELLAR (0.75−0.8, reported in ref. 7), which incorporates both molecular and spatial information (Fig. 7i, Supplementary Fig. 13d). In the HuBMAP dataset, the F1-score for TYPEx of 0.65 was lower than STELLAR (0.8, reported in ref. 7) and higher than Astir (0.46) and CELESTA (0.52) (Fig. 7j). Of note, while STELLAR showed the best performance based on the F1-score metrics, these scores were based on a model trained and tested within its original distribution. The performance of TYPEx could be affected by markers with low signal-to-noise ratio (Supplementary Fig. 14a−d) and different segmentation approaches (Supplementary Fig. 15).

In addition to the ground truth cell types, TYPEx also identified additional cell subpopulations distinguished by a unique combination of expressed protein markers. For example, in the CRC CODEX dataset, these included naive, cytotoxic and CD45RO memory CD8 T cells, and mature B cells, and cell states such as proliferation (Ki67$^+$), highly differentiated (CD57$^+$) and checkpoint molecule-expressing cells. In

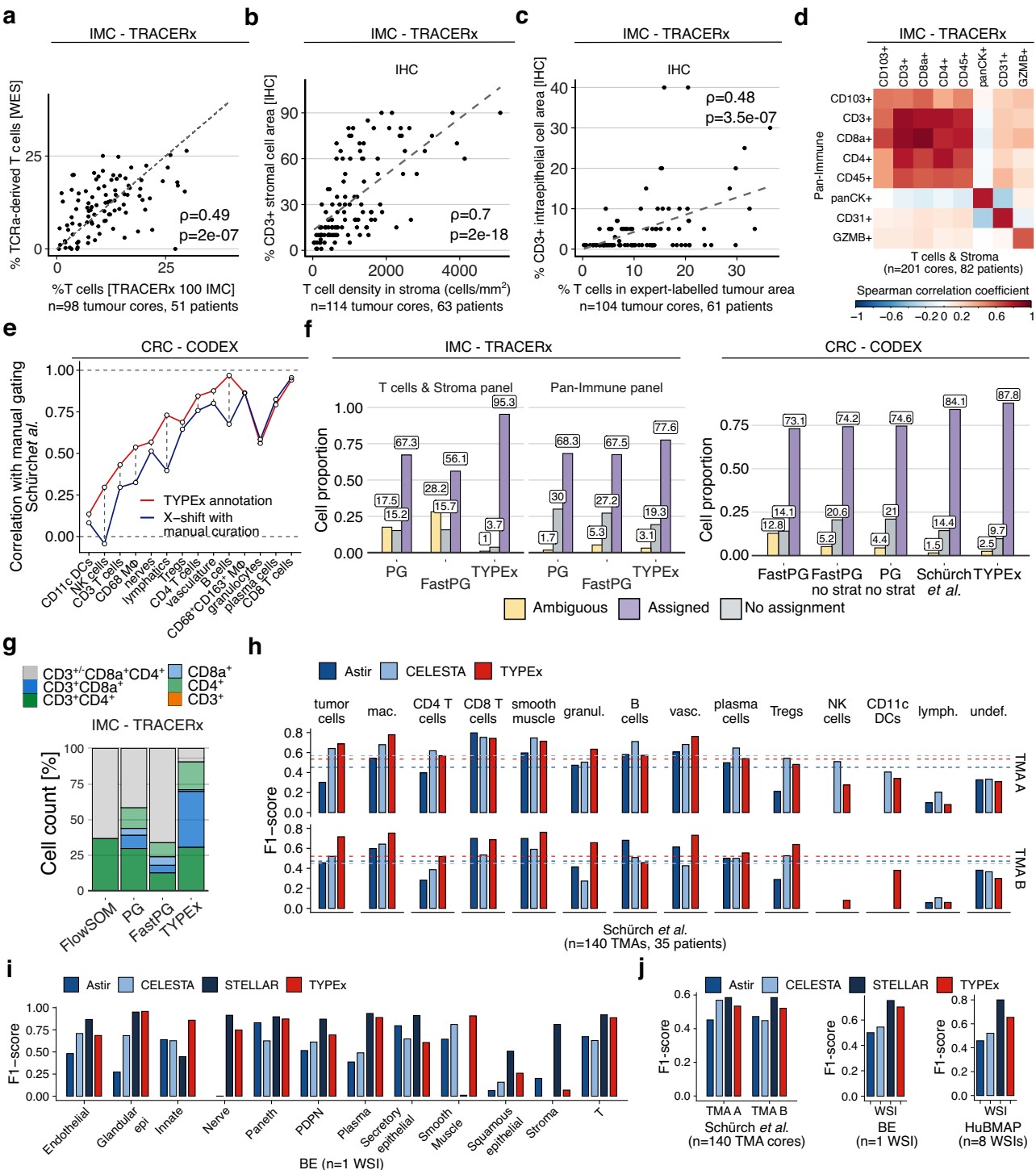

the BE dataset, TYPEx identified additional subpopulations of the manually annotated Innate (macrophages, neutrophils and dendritic cells) and T cells (CD4 and CD8 T cells) from the ground truth dataset (Supplementary Fig. 12a).

In summary, we provide a practical and automated approach with results comparable to manual curation and state-of-the-art cell phenotyping approaches.

## Discussion

Automating multiplexed imaging analysis without the need for high-level expertise is integral to improving reproducibility and accessibility in a growing experimental field. To address these needs, we present

PHLEX, a modular and user-friendly pipeline that provides solutions to key challenges in multiplexed imaging analysis. PHLEX integrates three modules specialising in different analytical tasks and performs comparably to similar standard approaches while providing complementary functionalities.

Addressing the need for an easy-to-use deep learning-based segmentation optimised for IMC, we developed the segmentation module deep-imcyto and benchmarked it against state-of-the-art deep learning segmentation approaches, confirming performance across different tissue types and fixation conditions[30–32]. While deep-imcyto is a fully automated pipeline, its use in *CellProfiler* mode requires prior manual configuration of the input files by the user. The cell phenotyping

**Fig. 7 | Evaluation of TYPEx performance in cell phenotyping and benchmarking against alternative approaches. a–c** Validation of TYPEx using orthogonal TRACERx data. Correlation of whole-exome sequencing (WES)-derived T-cell percentages calculated using T cell ExTRECT with the IMC-derived T-cell percentage on paired regional TMA tumour cores (**a**). Correlation of immunochemistry (IHC)-derived CD3[+] cells with the T-cell density in the stromal area (**b**) or the proportion of T cells over all cells in the intraepithelial area (**c**) detected from IMC on paired regional TMA cores (T cells & Stroma panel). **d** Comparison of cell densities calculated with TYPEx between two antibody panels from serial tissue sections within the TRACERx 100 cohort. The heatmap shows Spearman correlation coefficients between cell densities of markers and samples profiled with the Pan-Immune and T cells & Stroma panels. **e** Cell subtype annotations from TYPEx and published manually adjusted annotations from X-shift were correlated with the cell counts of annotations derived from manual gating in the CRC CODEX dataset (Spearman correlation). **f** Comparison of clustering approaches with TYPEx with

and without the stratification by confidence step (no strat). The fraction of Ambiguous and Unassigned cells in TRACERx 100 IMC (left) and CODEX (right) data were annotated according to the D score-derived positivity or published cell annotations (Schürch et al.). **g** The fraction of double-positive T cells derived from TYPEx compared to the three clustering approaches in the TRACERx 100 IMC cohort (n = 275 cores, T cells & Stroma panel). **h** Performance metrics F1-scores per cell subtype on the CRC CODEX dataset (n = 70 cores/TMA). CELESTA metrics on the TMA A cores were derived from a published confusion matrix (Zhang et al.). Dashed lines show the macro F1-scores across cell subtypes for each method. **i** Performance metrics F1-scores per cell subtype on Barrett's oesophagus (BE) CODEX data. **j**, Macro F1-scores for the three validation datasets. STELLAR scores were published previously (Brbić et al.). **a–c** ρ represents Spearman correlation coefficient with unadjusted p values. Source data are provided as a Source Data file. TILs tumour-infiltrating lymphocytes, H&E haematoxylin and eosin, CRC colorectal cancer, WSI whole-slide image.

module, TYPEx, performs automated yet comprehensive cell phenotyping and was benchmarked and validated using public CODEX imaging data from colorectal cancer, Barrett's oesophagus, and healthy intestine tissue. Together with the spatial analysis toolkit Spatial-PHLEX, these modules are packaged as a containerised Nextflow pipeline, supported with extensive documentation, and offer complementary features to comparable platforms[4].

TYPEx leverages existing methods and combines them with statistical approaches, providing additional functionalities to available cell phenotyping approaches (Table 1). TYPEx uses unsupervised clustering together with the repurposed probabilistic approach CellAssign[17] and implements the D-score to simultaneously determine the cell subtype and marker positivity. Compared to currently available methods (Table 1), TYPEx requires no manual assessment to define marker positivity, automatically resolves low-confidence calls and distinguishes unassigned and ambiguous cells. To detect marker positivity, TYPEx assumes that a marker is expressed on a subtype of cells. This assumption may not be fulfilled, for example, for β2-microglobulin, which is expressed on all nucleated cells except for a minority of tumour cells. In addition, the performance of TYPEx could be affected by high noise levels of markers, the segmentation approach, and the robustness of the clustering approach within a cell lineage and confidence group. Nevertheless, TYPEx allows for granular and robust phenotyping with accuracy comparable to manual gating[22], manually curated cell type annotations[7,22] and standard approaches[5–7]. While the performance of TYPEx was suboptimal for some rare cell subtypes due to lower marker specificity, TYPEx could capture both rare and frequent subpopulations with comparable accuracy. In addition, TYPEx incorporates tissue segmentation to quantify the cell phenotyping results as cell densities, which can be readily compared between different samples. TYPEx also provides a range of single-cell and pixel-level visualisations, summary graphs and figures to allow visual inspection and data exploration.

The user's expertise with a cell phenotyping tool likely affects its performance, as shown for CELESTA. We compared CELESTA between two TMAs comprising matched regions from the same tumours using the same parameters. CELESTA had higher performance scores on the TMAs for which the parameters were optimised by an expert user[5]. Similarly, for TYPEx, the validity of the user-provided cell type definitions can affect its performance.

While the deep learning-based tool STELLAR demonstrated the highest performance scores among the benchmarked cell phenotyping approaches, the factors that affect these scores, such as between-user variability, remain to be determined. As these scores were reported previously in STELLAR's study[7], the performance of STELLAR on independently trained models and datasets outside of its original distribution remain to be evaluated, in order to understand how it is affected by different antibody panels and clones, fixation conditions, image processing and segmentation approaches, noise and

multiplexed imaging modalities. Future work should seek to develop tools transferable across heterogeneous datasets and robust to noise. Combining deep learning-based with principled approaches can improve performance and transferability of each individual tool and accelerate and guide the training process, while facilitating data interpretation and discovery.

Spatial-PHLEX distinguishes itself from other spatial biology methods for cellular data through a combination of features, which we summarise in Table 2. Spatial-PHLEX provides a distinctive approach to tissue domain or cellular niche identification in spatial-omics data, specifically through density-based clustering of known cell types. Unlike other methods, this approach does not rely on a predefined window size or number of clusters to characterise cell niches (e.g. Spatial LDA[38], or spatial communities[22]). Instead, it takes into account purely the density of the cell type to be clustered and forms clusters from nearby cells only if the local density surpasses a threshold defined by the algorithm's *epsilon* parameter. Notably, the size of a cluster is solely determined by the density of the clustered cell type. Performing spatial clustering in this way may be useful where a particular cell type is of a priori interest and enables investigation of that cell type's organisation over different length scales and cluster sizes. This stands in contrast to methods that require post hoc inference of the major cell phenotype in an identified cluster or domain[22,38,39]. Spatial-PHLEX generates domain masks for downstream analysis, and conducts distinct domain measurements, for example, by calculating the distance of all other cells to cluster boundaries and generating intracluster densities and proportion measurements for all cell types. The capability of Spatial-PHLEX to uncover potential barriers between cell types is unique among spatial data analysis tools. These combined attributes position Spatial-PHLEX as a versatile tool with an integrated and feature-rich approach to spatial data analysis.

PHLEX was developed for reproducible analysis of large-scale IMC imaging projects and tested on Linux-based institutional clusters running SLURM and SGE task schedulers. Through its Nextflow implementation, PHLEX is adaptable to many different computer infrastructures. At present the Nextflow implementation of deep-imcyto requires a system with CUDA-enabled GPU hardware to run. However, we also provide py-imcyto, a streamlined Python repository for IMC segmentation that utilises the same trained model as deep-imcyto. Spatial-PHLEX barrier scoring algorithms are also GPU-accelerated due to their computational expense, and, thus, require the use of GPU hardware. However, Spatial-PHLEX may still be run in spatial clustering mode alone. Beyond basic knowledge of the command line to navigate to and edit the provided bash script for initiating Nextflow, programming skills and pathology expertise are not essential. Prior basic understanding of marker expression on different cell subtypes and lineages and any potential non-specific binding of antibodies are required for TYPEx configuration.

**Table 1 | Comparison of TYPEx with cell phenotyping approaches**

| Approach | Automated cell subtype assignment | Prior inputs | Automated detection of positivity/states | Tissue segmentation | Low-confidence calls | Visualisation and QC[a] | Summary outputs[b] | Dockerised |
|---|---|---|---|---|---|---|---|---|
| TYPEx | ✓ | cell type definitions | ✓ (cohort level) | ✓ | ✓ (automated) | ✓ | ✓ | ✓ |
| STELLAR[7] | ✓ | training data for the panel | ✓ (image and cohort level) | ✗ | ✗ | ✗ | ✗ | ✗ |
| Astir[6] | ✓ | cell type definitions | ✗, under development | ✗ | ✓ (requires user-defined threshold) | heatmap | ✗ | ✗ |
| CELESTA[5] | ✓ | cell type definitions | ✓ (optimisation, image and cohort level) | ✗ | ✓ (requires user-defined threshold) | scatter plot | ✗ | ✗ |
| FastPG[19] | ✗ | unsupervised | ✗ | ✗ | ✗ | ✗ | ✗ | ✓ |

[a]Marker intensity heatmaps and distribution plots, assigned cell object maps and scatter plots, overlays of marker intensity and positivity.
[b]Normalised cell counts (cell density), sample information, and spatial cell annotations. Summary outputs refer to summary tables such as normalised counts, summary quantifications with sample information, and spatial annotations of single-cell objects. QC quality control.

**Table 2 | Features of Spatial-PHLEX and other spatial methods for biological data**

| | Data type | Barrier scoring | Tissue domain identification through clustering | Density-based spatial clustering | Expression clustering | Cluster boundary determination and measurements | Cell type distance measurements | Neighbour interaction analysis | Nextflow implementation | Coding skills required |
|---|---|---|---|---|---|---|---|---|---|---|
| Spatial- PHLEX | Cell type | ✓ | ✓ | ✓ | ✗ | ✓ | ✓ | ✗ | ✓ | ✗ |
| UTAG[39] | Expression | ✗ | ✓ | ✗ | ✓ | ✗ | ✗ | ✗ | ✗ | ✓ |
| SpaGENE[45] | Expression | ✗ | ✗ | ✗ | ✗ | ✗ | ✗ | ✗ | ✗ | ✓ |
| Giotto[1] | Expression | ✗ | ✓ | ✗ | ✓ | ✗ | ✓ | ✓ | ✗ | ✓ |
| SpaGCN[46] | Expression | ✗ | ✓ | ✗ | ✓ | ✗ | ✗ | ✗ | ✗ | ✓* |
| Spatial-LDA[38] | Cell type | ✗ | ✓ | ✗ | ✗ | ✗ | ✗ | ✗ | ✗ | ✓ |
| Squidpy[43] | Both | ✗ | ✗ | ✗ | ✗ | ✗ | ✗ | ✓ | ✗ | ✓ |
| Scimap [scimap.xyz] | Cell type | ✗ | ✗ | ✗ | ✗ | ✗ | ✓ | ✓ | ✓ (through MCMICRO) | Some |
| NeighborhoodCoordination[22] | Cell type | ✗ | ✓ | ✗ | ✗ | ✗ | ✗ | ✗ | ✗ | ✓ |

* Ezy mode provided for new Python users

In conclusion, PHLEX provides an end-to-end computational framework for transforming raw multiplexed image data into interpretable quantitative cellular and spatial information, allowing complex analyses without high-level expertise. Offering a range of functionalities and systematically addressing the technical challenges of multiplexed image analysis, PHLEX has a broad potential user base and application in multiplexed spatial studies.

## Methods

### Population and subject details

TRACERx-PHLEX was developed using samples from prospectively recruited treatment-naive early-stage IA to IIIB non-small cell lung cancer (NSCLC) patients in the TRACERx study (https://clinicaltrials.gov/ct2/show/NCT01888601). The study was approved by an independent research ethics committee (13/LO/1546), the National Research Ethics Service (NRES) Committee London–Camden and Islington, with the sponsor's approval of the study by University College London (UCL/12/0279, project ID 138871)[10]. Mandatory informed consent for entry into the TRACERx study was obtained from every patient. Antibody panel development involved staining of positive and negative control lung, tonsil, cardiac and brain formalin-fixed paraffin-embedded (FFPE) tissues obtained from UCL/UCLH Biobank for Studying Health and Disease Renewal 2020 (ethics approval 20/YH/0088) and the PEACE study (ethics approval 13/LO/0972)[11]. These tonsil and additional kidney tissues sourced from the UCL/UCLH Biobank were further included as staining controls in the TRACERx IMC study.

TRACERx clinical samples analysed in this study were obtained from primary surgery. Snap frozen multi-region sampled tumour, lymph node, and adjacent normal tissues distant from the tumour within the resection specimen were processed to FFPE blocks after first taking sufficient material for DNA and RNA sequencing. A single representative 1.5 mm diameter core was punched from each regional FFPE block and assembled into tissue microarrays (TMAs). Pathology review of an H&E stain taken serially to the study sections resulted in reclassification of a subset of TRACERx patient tumour cores to benign tumour-adjacent due to a lack of tumour content. The TRACERx IMC cohort represents 83 patients, including 159 tumour cores, 21 benign tumour-adjacent cores, 1 lymph node metastasis core, 5 lymph node cores with no tumour content and 49 normal cores. Sex and ethnicity were not an inclusion or exclusion criteria in this study. One third of patients are females and two thirds of males; 96% of the cohort is from a white ethnic background and the mean age of the patients is 68, ranging between 34 and 85. See Supplementary Methods for panel-specific population details.

### TRACERx Nuclear IMC segmentation dataset

The TRACERx nuclear IMC segmentation dataset (TRACERx NISD) is a training dataset for nuclear instance segmentation in IMC. It comprises 41,962 expert-labelled nuclei across 116 256×256 μm image tiles. See Supplementary Methods for an overview of the dataset. We release the TRACERx NISD as a set of contrast-adjusted images with ground truth masks in 16-bit label format, as well as the binary masks of semantic features as used to train our segmentation model. Raw 41-channel IMC channel data corresponding to the NISD regions is available on request.

### deep-imcyto: nuclear and cell segmentation

**Nuclear segmentation: nuclear feature prediction.** The deep-imcyto nuclear segmentation model uses a UNet++ model trained on the TRACERx NISD to predict nuclear features which can be combined into instance-level nuclear masks (Fig. 2a). Similarly to a previously described approach[40], we trained our model to predict three properties of nuclear images: (i) nuclear material, (ii) the centre of mass of all nuclei and (iii) boundaries between touching nuclei, using a hybrid loss

function consisting of an even weighting of Dice loss and binary cross entropy. As the UNet++ model contains a large number of trainable parameters ($n = 36,157,893$), prior to training on the TRACERx NISD, we first trained the model using nuclear masks produced via a classical (CellProfiler[15]) segmentation of our IMC images. As others have found[41], this strategy aids final model convergence through the generation of initialisation weights sufficient for transfer learning with the smaller scale expert-labelled data. We also performed extensive augmentation during model training (Supplementary Methods).

During training, we negatively weighted the edge of nuclear material masks to facilitate improved instance predictions in crowded cell regions. We then converted our semantic feature predictions into instance masks by combining nuclear material and boundary predictions into catchment basins and using thresholded versions of the centre of mass and nuclear material predictions, to perform a marker-controlled watershed. Preliminary nucleus instance masks then underwent a series of post-processing steps. Firstly, small object removal was performed to discard masks under a size threshold (<4 pixels), and subsequently an instance-level binary morphological closing operation was applied to remove internal crack-like artefacts in each nuclear mask and improve the smoothness of the mask's perimeter. The effect of this instance morphological closing versus standard grayscale closing is shown for a nuclear label mask with simulated crack artefacts in Supplementary Fig. 2a.

**Nuclear segmentation: error correction model.** To determine cases where lower confidence centre-of-mass predictions should split larger but morphologically improbable nuclear masks, we designed a multi-stage watershed procedure guided by error prediction. To do this, we trained an anomaly detection model comprising a lightweight fully connected autoencoder of 524 trainable parameters from an architecture of (15, 7, 3, 7, 15) neurons per layer with ReLu activation and dropout (0.1) after each dense layer. Input for the model is a vector of the following 15 nuclear morphometric measurements: nuclear area; eccentricity; Euler number; extent; maximum Feret diameter; the seven Hu moments; nuclear perimeter; Crofton perimeter and solidity. We derived these measurements for all ground truth nuclei in the TRACERx NISD and scaled each to zero-mean and unit variance prior to training. We trained the autoencoder model on the scaled ground truth morphometry data with a batch size of 256 for 100 epochs with the Adam optimizer and mean-squared logarithmic error loss function, allowing early stopping after 10 consecutive epochs of no improvement. All models were written in Python 3 with Keras and Tensorflow 2.

We applied our trained model to the same features extracted from the predicted nuclear masks and used the mean-squared error from the autoencoder model for each predicted mask to flag masks with error rates above a threshold, corresponding to morphologically suspect nuclei. For these high-error nuclei, marker-controlled watershed was performed again on predicted feature images, but this time with a lower intensity threshold in the centre-of-mass (i.e. marker) image, enabling lower confidence nuclear centroids to be picked up. Corrected nuclear masks were then post-processed as already described (small object removal; morphological closing) before this process of morphometric measurement and detection and re-segmentation of high-error nuclei was repeated. Any nucleus masks with errors over the error threshold at this stage were removed entirely from the refined labels and then, pixels of erroneous masks in the (post-processed) first iteration watershed were replaced with the corresponding pixels from the second iteration watershed (adjusted to ensure unique labels in the resultant image) (Supplementary Fig. 2b). In our implementation we used an error threshold of 1.0, corresponding to nuclei with prediction errors within roughly the top 2.5% of all predictions. Morphometry correction of 7917 anomalous nuclei out of the 2,996,197 (0.26%) identified in images from our T cells & Stroma IMC panel resulted in 17,897 nuclei with size and shape metrics more closely resembling the

ground truth distribution than was observed pre-correction (Supplementary Fig. 2c).

**Multiplexed consensus cell segmentation (MCCS).** To segment whole cells in images from the TRACERx lung dataset, we designed and implemented a whole-cell segmentation method called Multiplexed Consensus Cell Segmentation (MCCS) as a particular implementation of the *CellProfiler* mode of deep-imcyto.

First, a set of whole-cell masks was created by applying Otsu thresholding to determine foreground in each of a user-curated set of cell lineage marker ("segmentation marker") images which had undergone minimal preprocessing (spillover compensation, hot pixel removal, median filtering window size = 3) (Supplementary Methods, Supplementary Table 4). Deep-learning nuclei from deep-imcyto were then tested for overlap with each of these segmentation marker masks in turn. Nuclei associated with a particular segmentation marker were then used as seeds for segmentation marker-specific cell identification using propagation onto the relevant minimally preprocessed segmentation marker image. A consensus mask was then generated by combining each of the individual segmentation marker masks such that all pixels that ended up assigned to the same single cell must have been assigned as cell foreground in the same set of segmentation marker cell masks. In our implementation for lung cancer tissue cores, MCCS included an additional step to capture frequently observed non-nucleated αSMA⁺ fibroblasts by implementing primary object identification on a preprocessed αSMA channel. These cells were subsequently combined with the nucleated cell mask to produce a final whole-cell segmentation harbouring both populations. Finally, size criteria (minimum cell size 9 pixels, maximum cell size 1000 pixels) were implemented to represent the expected range for such imaged features, generating a final whole-cell mask. Single cell measurements of all IMC marker mean intensities were then extracted and input into the TYPEx multiplexed cell phenotyping module. Selection of thresholding, overlap and size criteria for MCCS segmentation is described in Supplementary Methods. MCCS was implemented using CellProfiler v3.1.9[15].

**Pseudo-H&E creation.** We developed a standalone Python 3 library for generating pseudo-coloured H&Es from multiplexed image data within deep-imcyto. Raw data for DNA channels were contrast-adjusted and denoised, and the set of IMC channels to be incorporated into the H&E counterstain were combined through an average intensity projection to form a counterstain image. This counterstain image was contrast-adjusted and denoised, then custom colour maps were applied to the grayscale nuclear and counterstain images before recombination into the final "pseudo-H&E" (Fig. 2a). Pseudo-H&Es can be generated for any type of multiplexed imaging data, including co-detection by indexing (CODEX) data, provided the appropriate channels are provided to the software.

**Validation and benchmarking of deep-imcyto**
**Other segmentation models.** For Mesmer and Cellpose prediction, two channel images were created for input to the models, with an average of nuclear marker channels in the first channel and a mean intensity projection of all other markers in our IMC panel in the second. We used the Mesmer implementation from the deepcell-tf Python library to perform all predictions with default settings. Default parameters for Cellpose did not perform well on our IMC data, necessitating an increase in the flow_threshold parameter to 1. Pretrained Stardist models were used with default prediction parameters. We retrained the Stardist model on the TRACERx NISD for 200 epochs with default training parameters.

**Segmentation scoring.** All segmentation scoring metrics unless otherwise specified were calculated using the deepcell-toolbox Python library. $n_{predict}/n_{gt}$ *objects* was calculated as the ratio of predicted objects to the number of ground truth objects in a given image. *Bijective cardinality* (the number of labelled objects in the ground truth dataset with only a single overlapping label in the predicted dataset) and *segmentation similarity measure* were calculated according to ref. [41] using a custom Python script.

**TYPEx: cell phenotyping**
For the analysis of TRACERx IMC data, TYPEx used the mean marker intensities per segmented cell object in the format of a cell-by-marker intensity matrix. The input matrix was generated by deep-imcyto in *CellProfiler* mode using the MCCS procedure. For the analysis of the colorectal cancer (CRC) CODEX dataset, the intensities and coordinates for each cell object were obtained from the Mendeley Data repository as provided by Schürch et al. [22]. The cell objects and the expert-labelled annotations of CODEX data on intestine (HuBMAP) and Barrett's oesophagus (BE) were downloaded from the online repository Dryad as provided in the original publication[7].

The cell-type definitions file includes a list of cell lineages/subtypes and a corresponding set of marker proteins that identify each cell lineage. The set of marker proteins are assumed to have a higher expression in the cell types they define. We defined 13 major cell lineages targeted by our two antibody panels to ensure that each cell in the cohort can be covered by a major cell lineage definition. For the T cells & Stroma panel, we defined the following major cell lineages with the major cell lineage markers in brackets: Vimentin+ cells (Vimentin), Endothelial cells (Vimentin, CD31), Epithelial cells (pancytokeratin/panCK), αSMA⁺ cells (Vimentin, αSMA), CD4 T cells (CD45, CD3, CD4), CD8 T cells (CD45, CD3, CD8a), T cells - Other (CD45, CD3), Leukocytes - Other (CD45). The major cell lineages for the Pan-Immune panel were Endothelial cells (CD31), Epithelial cells (panCK), CD4 T cells (CD45, CD3, CD4), CD8 T cells (CD45, CD3, CD8a), T cells - Other (CD45, CD3), B cells (CD45, CD79a, CD20), Monocytes (CD45, CD11b, CD14), Macrophages (CD45, CD11b, CD14, CD68, CD206, CD163), Myeloid cells - Other (CD45, CD11b), Leukocytes - Other (CD45). Some markers, such as CD45 and vimentin, are expressed by multiple cell lineages. These shared proteins are used to infer a hierarchy of cell lineages. This hierarchy based on cell subtype and lineage definitions is considered for both cell stratification and annotation. The cell subtype definitions are provided with the GitHub distribution of TYPEx in the cell type annotation config file.

To stratify by major cell lineage, TYPEx used the probabilistic approach CellAssign[17], setting the cell-specific size factor to one. For batch effect correction, we specified a unique TMA identifier for the TRACERx and CODEX datasets. CellAssign was run with the following default parameters indicated in the config file *typing params*: *learning rate* = 0.01 and *shrinkage* = true. CellAssign was configured to allocate the most likely cell type for each cell, even in cases of low probability scores using the parameter *assign_prob*.

To differentiate between low and high-confidence assignments, the probabilistic model is provided with partial information in order to identify the cell annotations that remain stable and those that vary compared to the model with complete information. In the TRACERx IMC dataset on the T cells & Stroma panel, to generate training labels for low- and high-confidence assignment, we excluded Mesenchymal cells/Vim+ cells from the cell lineage definitions input file. Mesenchymal/Vim+ cells were defined as those expressing vimentin alone and were negative for other lineage-specific markers defining other vimentin-expressing cells, such as αSMA⁺ cells and immune cells. The associated marker vimentin was also excluded from the input cell-by-marker matrix. This guarantees that no other marker in the remaining input list can correctly identify the excluded cell lineage, and any new assignment will be considered a varying/low-confidence assignment. For αSMA⁺ cells, for example, which are defined to be expressing vimentin and αSMA in the complete model, only αSMA will be

considered in the incomplete model. Using a binomial logistic regression model to distinguish between stable and varying annotations by their probability score and mean intensity, TYPEx predicts and classifies low and high-confidence cells.

For TRACERx analysis, where deep-imcyto was used, the cell-marker-specific segmentation masks from the MCCS approach were considered to further filter the stable labels as likely true calls. We overlaid segmented cell objects from the final cell object masks with the segmented objects from the marker-specific cell object masks. When the final cell object overlapped with a cell object from the marker-specific segmentation of a cell-lineage-specific marker, the cell object was assigned the lineage defined by the marker. The overlap was determined based on centroid separation by less than 5 microns. The remaining objects were considered undetermined. The likely true objects were considered the stable annotations from the probabilistic model that matched the annotation from the marker-specific cell object masks. The likely-false objects were considered the variable annotations from the probabilistic model that mismatched the annotation from the marker-specific cell object masks. The binomial logistic regression model for the TRACERx datastets was built to best separate the likely true and likely-false cell type calls, and then applied on all objects (including undetermined) to predict and classify them as high- or low-confidence cell type calls.

For cell clustering within a major cell lineage and confidence group, the following parameters were used as defined in the config file *typing params* for each method: $k = 30$ for FastPG[19] and Rphenograph[18], *scale* = false, *maxClust* = 80, and *rlen* = 10,000 for FlowSOM[42], and no transformation was applied to the raw mean pixel intensities per cell prior to clustering.

To perform automated marker detection, TYPEx relies on a subset of co-expressed and mutually exclusive markers in the antibody panel. In the TRACERx IMC and the CRC CODEX data analysis, we used CD3, CD4 and CD8. We specified $CD3^+CD4^+$ and $CD3^+CD8a^+$ cells as high frequency populations, and $CD8a^+$ ($CD3^-$), $CD4^+CD8a^+$, $CD3^+CD4^+CD8a^+$ cells as unexpected/rare populations. Finally, $CD3^+$ cells were expected with low frequency while $CD4^+$ cells were expected with variable frequency.

In the Barretts's oesophagus CODEX dataset, due to high CD8 intensities on the myeloid cell populations (CD11b+ and CD15+), single-positive CD8 were excluded from the list of rare subpopulations and single-positive CD4 was added instead. This model with CD8 added to the Major cell lineage took longer compared to the HuBMAP dataset, which has a comparable number of cells and major cell lineage definitions (Supplementary Table 3).

## TYPEx: tissue segmentation
We trained a three-class classifier for Background, Tumour/Epithelium and Stroma on composite images from NSCLC tumour, tumour-adjacent lung tissue and lymph nodes using ilastik[20]. We selected DNA intercalators, markers with tissue-specific expression for the epithelium (pancytokeratin) and stroma (immune-specific biomarkers, αSMA, CD31) as input features (Supplementary Table 5, Fig. 3e). Pre- and post-processing of the images is automated without the need for manual image-specific adjustments. When available, vimentin (T cells & Stroma panel), collagen1 (T cells & Stroma panel) and panactin (Pan-Immune panel) were also included in areas with mutual exclusivity with the epithelial cell markers. The performance of this classifier was validated through pathology review of paired H&E images.

## Reproducibility and validation of TYPEx
**Raw intensity distributions by cell types and positivity status.** The raw intensity distributions of cell type-sepecific markers between different cell types were illustrated for each marker across all evaluated cell subtypes, taking the median cellular pixel intensity per image (Supplementary Fig. 16). The cellular pixel intensity is the mean intensity of the pixels within a cell boundary. Here, we plotted low and high-confidence cell type calls separately for 11 cell-type-specific indicator markers across the T cells & Stroma antibody panel, demonstrating that the intensity of the indicator marker was higher in the cells assigned with the cell subtype associated with that indicator marker.

**Comparison between antibody panels.** We compared the two antibody panels, Pan-Immune panel and T cells & Stroma panel, by the cell densities of major cell lineages and cell types (Supplementary Fig. 10d) demonstrating reproducibility of the cell assignment between the panels.

**Clustering reproducibility analysis.** The total number of cells analysed were 3,157,972 for antibody panel T cells & Stroma and 3,168,689 for Pan-Immune panel. We subsampled 2/3 of the cells in the dataset and analysed the subset with CellAssign and the three clustering approaches implemented in TYPEx (Supplementary Fig. 13a, b). Similar results were obtained when subsampling 2/3 of the cells per image. For cluster comparison, we matched the statistically most similar clusters, and the percentage of cells between matched clusters were compared. For comparison by major cell lineage, the clusters were annotated based on positivity called according to the D-score approach and assigned based on the provided major cell lineage definitions.

**Comparison with orthogonal TRACERx data.** Publicly available tumour-infiltrating lymphocyte (TIL) scores derived from pathologist evaluation of regional tumour H&E slides were correlated with IMC-derived TIL densities calculated as the T cells and B cells count per tissue area in TMA cores [mm$^{-2}$]. TIL scores represented the area occupied by mononuclear inflammatory cells over the total viable intratumoral and stromal areas. Cancer cell area, TLS and lymphoid aggregate area, and necrotic area were excluded from assessment.

Stroma and tumour-localised T-cell densities and percentages derived from IMC data (T cells & Stroma IMC panel) were compared with available paired anti-CD3 immunohistochemistry (IHC) data. The percent $CD3^+$ T cells localised to viable stroma and intratumour areas were scored by a pathologist.

Comparisons with flow cytometry data were restricted to a subset of 17 tumour regions with a comparable fraction of assayed T cells. The 17 tumour cores were selected as they had at least 2000 T cells identified by IMC, which matched the minimum number of T cells evaluable by flow cytometry. T-cell phenotypes assessed included $CD8^+CD103^+$ T cells, $CD57^+CD8^+$ T cells, $CD27^+CD8^+$ T cells, $CXCR6^+CD8^+$ T cells, and $CD45RA^+CD8^+$ T cells. The percentage of T cells derived from flow cytometry was compared to the percentage of T cells derived from IMC.

Publicly available TCRA scores derived from WES data using the T cell ExTRECT tool[34] were correlated with paired T-cell percentages derived from whole IMC images.

**Validation with public CODEX datasets.** Publicly available CODEX imaging data of 140 FFPE TMA cores from 35 colorectal cancer patients was used to evaluate the cell phenotyping performance of TYPEx. The manual gating data was downloaded from the CellEngine experiment provided with the original publication[22].

For the CRC CODEX dataset, the cell type definitions file was designed to match the published annotation names[22]. This analysis was done independently of deep-imcyto. Therefore, tissue segmentation was not considered in cell assignment. Ambiguous/Unassigned cells included cells classified as tumour/immune cells, immune cells or stroma in the previously published annotations. $CD11b^+$ monocytes and granulocytes were combined for comparison with X-shift.

For the Barrett's oesophagus (BE) and the HuBMAP CODEX datasets, the cell subtypes were defined to match the definitions of the

ground truth annotations based on the predicted marker expression in the original study[7]. A heatmap of mean marker intensities across the defined cell subtypes for the BE dataset is illustrated in Supplementary Fig. 12b.

Before being input to Astir, the intensity matrices of the three CODEX validation datasets (CRC CODEX, BE and HuBMAP) were normalised with arcsinh transformation and cofactor 150 and winsorsised to 0–99% for outlier removal[6]. The markers used by TYPEx to assign cell types were included in yaml format as input for Astir.

CELESTA[5] was run using the cell type definitions provided with the GitHub distribution of the CRC CODEX dataset (Schürch et al.). The anchor and iteration thresholds optimised for the test dataset TMA A provided with the GitHub release were used for the analysis of TMA B from the CRC dataset. For the validation datasets BE and HuBMAP B004, CELESTA was run with the parameter *transform_type* set to false, as the downloaded intensity matrices were previously normalised. The high anchor thresholds were automatically set to the 25th percentile of the probabilities for the true positive cell type in the ground truth dataset. The low anchor thresholds were set to 0.95 for all cell types. The index thresholds were set to be higher than the corresponding anchor threshold by 0.05. The markers used to assign cell types by TYPEx were reformatted in the required input for CELESTA. However, including any of the markers Vimentin, CD138, CD11c and p63 in the cell type definitions resulted in NAs for all assignment probabilities; therefore, these were excluded from the input prior marker information file.

### Spatial-PHLEX: spatial clustering
In the Spatial-PHLEX spatial clustering workflow, coordinate data corresponding to a specified cell type (e.g., epithelial cells) is clustered into local regions of spatially associated cells with the DBSCAN algorithm[27]. Cells are grouped in the same cluster where their local density exceeds that allowed by the *epsilon* (EPS) clustering parameter (set in the Spatial-PHLEX config). Once spatial clusters of the specified cell type have been established, the boundaries of all clusters are determined using the alpha shape algorithm. Using a computational geometry approach with the Python library Shapely, we then determine the cells of all other phenotypes that reside within the boundaries of each spatial cluster.

Spatial-PHLEX performs clustering for all unique cell types present in the coordinate data and parallelises this process across images. For a given dataset, an appropriate EPS parameter and alpha radius value must be established by the user; inappropriate values may produce poor results depending on tissue type and overall cell densities.

### Spatial-PHLEX: cellular barrier scoring
The barrier scoring workflow of Spatial-PHLEX first creates the cell spatial graph from cell coordinate data using Squidpy[43]. The barrier module then performs a shortest paths analysis from each cell of the source type (e.g., CD8 T cells) to the target cell type (e.g., tumour cells), and resolves the cellular phenotypes of all nodes along this path. The module utilises a CUDA-accelerated single shortest paths algorithm[44] to find the paths to all other graph nodes from the starting cell. These paths are then refined to obtain the shortest paths to the target cell type. Within Spatial-PHLEX, barrier scoring can be run in conjunction with spatial clustering, for example, to restrict the barrier score to agglomerations of the target cell type larger than a set threshold. We devised two scores for quantifying the degree and location of barrier cells along a given cell path, for which we provide a visual representation in Supplementary Fig. 7b. These scores fall into two categories: [1] a score that quantifies the overall number of barrier cells along paths to tumour cells, and [2] a score which quantifies the number of cells which are adjacent to tumour cells. Furthermore, there may be multiple cells of the target type at the shortest path length, and to account for this we introduce so-called "all-paths" barrier scores

which for a given source cell are the average barrier content over all paths of the same length that end in the target cell type. Sample-level summary scores are calculated as the mean, median and standard deviation of the barrier scores over all source cells.

### Reporting summary
Further information on research design is available in the Nature Portfolio Reporting Summary linked to this article.

## Data availability
The TRACERx Nuclear IMC segmentation dataset and deep-imcyto's trained neural network model weights can be downloaded from Zenodo under accession code 7973724. A testing dataset consisting of five samples from the TRACERx IMC data (T cells & Stroma panel) has been deposited on Zenodo under accession code 7973724 and the TRACERx-PHLEX GitHub repository. The raw IMC data generated in Enfield et al.[11] and analysed in this study are available under restricted access due to privacy through the CRUK and UCL Cancer Trials Centre (ctc.tracerx@ucl.ac.uk) for academic non-commercial research purposes. Access will be granted upon review of a project proposal, which will be evaluated by a TRACERx data access committee and entered into an appropriate data access agreement, subject to any applicable ethical approvals. WES data from the TRACERx cohort included in this study have been deposited at the European Genome–phenome Archive (EGA) with details on how to apply for access. The data are hosted by The European Bioinformatics Institute and the Centre for Genomic Regulation (CRG) under the accession code EGAS00001006494 (WES); access is controlled by the TRACERx data access committee due to the nature of the data and commercial partnerships arrangements. For the publicly available colorectal cancer CODEX dataset, the cell-by-marker intensity table and the published cell-type annotations were available under https://doi.org/10.17632/mpjzbtfgfr.1, and the manual gating information was obtained from CellEngine under accession number 5ea1170788ae4203c2959042. The CODEX datasets on healthy intestine (HuBMAP) and Barrett's oesophagus (BE) together with the associated ground truth annotations were provided on the online repository Dryad https://datadryad.org/stash/share/1OQtxew0Unh3iAdP-ELew-ctwuPTBz6Oy8uuyxqliZk. Raw tiff files used in this study for validation and benchmarking of the segmentation approach are available on a Figshare repository under the handle 10779/crick.c.5270621.v2 for the mouse IMC data from lung cancer, on Zenodo under accession code 3518284 for the human breast cancer IMC data, and on datasets.deepcell.org for TissueNet breast cancer data. The data generated and visualised in this study are provided as Source Data files. Source data are provided with this paper.

## Code availability
The TRACERx-PHLEX pipeline is available on GitHub along with instructions on how to run it on the PHLEX test dataset: https://github.com/FrancisCrickInstitute/TRACERx-PHLEX. All software dependencies were integrated in the Nextflow pipeline and deposited as containers on public dockerhub repositories (magnesa, mihangelova, quay.io, cellprofiler, and ilastik). The deep-imcyto core nuclear prediction model is available for users who wish to use it outside of the deep-imcyto Nextflow pipeline: https://github.com/FrancisCrickInstitute/py-imcyto. The PHLEX online documentation includes user guides and additional technical details on PHLEX functionalities and potential use cases, available on https://tracerx-phlex.readthedocs.io/.

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

## Acknowledgements

We are grateful for assistance from the Experimental Histopathology, Flow Cytometry, and Scientific Computing facilities at the Francis Crick

Institute. The TRACERx study (Clinicaltrials.gov no: NCT01888601) is sponsored by University College London (UCL/12/0279) and has been approved by an independent Research Ethics Committee (13/LO/1546). TRACERx is funded by Cancer Research UK (C11496/A17786) and coordinated through the Cancer Research UK and UCL Cancer Trials Centre, which has a core grant from CRUK (C444/A15953). We gratefully acknowledge the patients and relatives who participated in TRACERx and PEACE studies. We thank all site personnel, investigators, funders and industry partners who supported the generation of the data within this study. This work was supported by the Francis Crick Institute, which receives its core funding from Cancer Research UK (CC2041), the UK Medical Research Council (CC2041), and the Wellcome Trust (CC2041). This work was also supported by the Cancer Research UK Lung Cancer Centre of Excellence and the CRUK City of London Centre Award (C7893/A26233), as well as the UCL Experimental Cancer Medicine Centre. This work was supported by funding as part of a research collaboration with Bristol Myers Squibb. This project has received funding from the European Research Council (ERC) under the European Union's Horizon 2020 research and innovation programme (grant agreement no. 101018670). For the purpose of Open Access, the author has applied a CC BY public copyright licence to any Author Accepted Manuscript version arising from this submission. C.S. is a Royal Society Napier Research Professor (RSRP\R\210001). His work is supported by the Francis Crick Institute, which receives its core funding from Cancer Research UK (CC2041), the UK Medical Research Council (CC2041), and the Wellcome Trust (CC2041). C.S. is funded by Cancer Research UK (TRACERx (C11496/A17786), PEACE (C416/A21999) and CRUK Cancer Immunotherapy Catalyst Network); Cancer Research UK Lung Cancer Centre of Excellence (C11496/A30025); the Rosetrees Trust, Butterfield and Stoneygate Trusts; NovoNordisk Foundation (ID16584); Royal Society Professorship Enhancement Award (RP/EA/180007 & RF\ERE \231118); National Institute for Health Research (NIHR) University College London Hospitals Biomedical Research Centre; the Cancer Research UK-University College London Centre; Experimental Cancer Medicine Centre; the Breast Cancer Research Foundation (US; BCRF-22-157); Cancer Research UK Early Detection and Diagnosis Primer Award (Grant EDDPMA-Nov21/100034); and The Mark Foundation for Cancer Research Aspire Award (Grant 21-029-ASP) and ASPIRE II award (23-034-ASP). This work was supported by a Stand Up To Cancer-LUNGevity-American Lung Association Lung Cancer Interception Dream Team Translational Research Grant (Grant Number: SU2C-AACR-DT23-17 to S.M. Dubinett and A.E. Spira). The indicated Stand Up To Cancer grant is administered by the American Association for Cancer Research, the Scientific Partner of SU2C. C.S. is in receipt of an ERC Advanced Grant (PROTEUS) from the European Research Council under the European Union's Horizon 2020 research and innovation programme (grant agreement no. 835297). E.C. and E.S. were partly funded by The Mark Foundation for Cancer Research (MFCR ASPIRE 2022-0384). E.S. is additionally supported by the European Research Council (ERC Advanced Grant CAN_ORGANISE, Grant agreement number 101019366) and the Francis Crick Institute, which receives its core funding from Cancer Research UK (CC2040), the UK Medical Research Council (CC2040), and the Wellcome Trust (CC2040). F.vM. is the recipient of the Amsterdam UMC fellowship and receives funding from the European Research Council under the European Union's Horizon Europe WIDERA work programme (grant agreement No. 101079113). K.S.S.E. was supported by the European Union's Horizon 2020 research and innovation programme under the Marie Skłodowska-Curie grant agreement No. 838540 and is supported by the Royal Society (RF\ERE\210216). M.A. was supported by the Momentum Fellowship from The Mark Foundation for Cancer Research (24-008-FSH).

## Author contributions

A.M., E.C., K.S.S.E., C.S. and M.A. designed the study. A.M., E.C. and M.A. conceived the methodology. A.M. and M.A. wrote the code. K.S.S.E, D.A.M., M.Sivakumar, D.L., P.S.H. and J.L.R. performed the experiments and generated the data. A.M., E.C., K.S.S.E., C.L., M.Shimato, E.D. and M.A. performed analyses and data curation. A.M., E.C., and M.A. performed data validation. K.V., D.L., C.T.H., P.S.H., F.vM., J.L.R. and S.A.Q. provided technical support and conceptual advice. A.M., E.C., K.S.S.E., C.L., and M.A. generated the figures. A.M., E.C., K.S.S.E., C.S. and M.A. wrote the original draft. A.M., E.C., K.S.S.E., C.L., J.D., E.S., C.S. and M.A. reviewed and edited the manuscript.

## Funding

## Competing interests

C.S. acknowledges grant support from Bristol Myers Squibb related to this work and grants from AstraZeneca, Boehringer-Ingelheim, Pfizer, Roche-Ventana, Invitae, Ono Pharmaceutical, and Personalis outside of the submitted work. He is Chief Investigator for the AZ MeRmaiD 1 and 2 clinical trials and is the Steering Committee Chair. He is also Co-Chief Investigator of the NHS Galleri trial funded by GRAIL and a paid member of GRAIL's Scientific Advisory Board. During the conduct of the study outside the submitted work, C.S. has received consultant fees from Achilles Therapeutics (also a SAB member), Bicycle Therapeutics (also a SAB member), Genentech, Medicxi, China Innovation Centre of Roche (CICoR) formerly Roche Innovation Centre—Shanghai, Metabomed (until July 2022), Relay Therapeutics SAB member, Saga Diagnostics SAB member and the Sarah Cannon Research Institute. Outside of the submitted work, C.S. has received honoraria from Amgen, AstraZeneca, Bristol Myers Squibb, GlaxoSmithKline, Illumina, MSD, Novartis, Pfizer, Medixci, and Roche-Ventana. C.S. has previously held stock options in Apogen Biotechnologies and GRAIL, and currently has stock options in Epic Bioscience, Bicycle Therapeutics, Relay Therapeutics, and has stock options and is co-founder of Achilles Therapeutics. C.S. declares a patent application (PCT/US2017/028013) for methods for lung cancer; targeting neoantigens (PCT/EP2016/059401); identifying patient response to immune checkpoint blockade (PCT/EP2016/071471); methods for lung cancer detection (US20190106751A1); identifying patients who respond to cancer treatment (PCT/GB2018/051912); determining HLA LOH (PCT/GB2018/052004); predicting survival rates of patients with cancer (PCT/GB2020/050221); methods for systems and tumour monitoring (PCT/EP2022/077987). C.S. is an inventor on a European patent application (PCT/GB2017/053289) relating to assay technology to detect tumour recurrence. This patent has been licensed to a commercial entity and under their terms of employment C.S. is due a revenue share of any revenue generated from such license(s). M.A. is a co-inventor on a European patent application (PCT/EP2020/059272) about methods for predicting and preventing cancer in patients with premalignant lesions. J.D. reports grants, personal fees, and nonfinancial support from AstraZeneca, personal fees from Bayer, Jubilant, Theras, Bridge-Bio, Vividion, Novartis, and grants and nonfinancial support from Bristol Myers Squibb and Revolution Medicines, outside the submitted work. E.S. has received funded research agreements from Merck Sharp Dohme, AstraZeneca and personal fees from Phenomic outside the submitted work. C.T.H. has received speaker fees from AstraZeneca, holds a paid advisory role for GenesisCare UK, research funding and support from Roche, AstraZeneca and Personalis. S.A.Q. reports other support from Achilles Therapeutics, grants from Roche, and Sairoopa outside the submitted work. J.L.R. reports speaker fees from Boehringer Ingelheim and GlaxoSmithKline, consults for Achilles Therapeutics Ltd and has filed patents for cancer early detection (PCT/EP2023/076521 and PCT/EP2023/076511). D.A.M. reports speaker fees from AstraZeneca and Takeda, consultancy fees from AstraZeneca, Thermo Fisher, Takeda, Amgen, Janssen, MIM Software, Bristol Myers Squibb and Eli Lilly and educational support from Takeda and Amgen. The remaining authors declare no competing interests.
