## [Peer Review File · Nature Communications]

Reviewers' Comments:

Reviewer #2:

Remarks to the Author:

The authors have sufficiently addressed my concerns.

Just one minor comment that Extended Data Figure 4 is cut.

Please see below comments regarding the points raised by reviewer #1:

1. Regarding concerns on the generalization of the method beyond TMAs to CODEX data: The analysis presented is a step in the right direction towards benchmarking the methods but it is not done systematically enough. Why was only one patient used in the CODEX HuBMAP dataset (B004, Figure Extended Data Figure 8-9)? Why wasn't precision and recall measured using CELESTA in the CODEX/HuBMAP dataset? The authors should run all competing approaches (including trying Astir again) in both IMC and CODEX datasets, and additional datasets with ground truth.
2. Thank you for addressing the concerns regarding computational complexity/time.
3. The use of a Z-score in Figure 6b makes the range of the metrics hard to interpret, particularly given that many have nicely bounded ranges. Furthermore, for a few metrics such as "merge", "missed_det_from_merge", "true_det_in_catastrophe", "pred_det_in_catastrophe", "catastrophe", CellPose (nuclear) seems to perform better than deep-imcyto. The authors should mention on the Methods how these metrics are calculated and comment on potential reasons for worst performance.
4. The authors "acknowledge [...] the subjective and time-consuming nature of manual review" involved in the selection of segmentation markers. However, this seems to now only be highlighted in the Tutorial and not in the manuscript. In the current version of the manuscript, TRACERx-PHLEX is still labeled as an "automated" pipeline. Furthermore, the previous reviewer comments regarding "the rationale to choose filtering parameters" not being explained was not incorporated in the manuscript and should be present in the Methods section.
5. Thank you for the explanation regarding scaling of the distributions. Please make this section clearer in the Methods section as well and not only in the Tutorial so the approach can be reproduced outside of the code provided by the authors as well.
6. Thank you for the explanation
7. Thank you for detailing the rationale in more detail.
8. Thank you for acknowledging the limitations. Please mention them in the manuscript (e.g. Discussion section), and not only in the Tutorial.
9. I appreciate the effort made by the authors to illustrate the effect of threshold decision, however the original reviewer comment requested to "demonstrate this quantitatively". Can the authors for example simulate different levels of noise to real-world datasets and assess the effect variance in threshold estimate has on the abundance of populations/positive populations (e.g. with correlation).
10. Thank you for the explanation. However, the strategy to use the intensity range of some markers (CD3, CD4 and CD8a) to determine the threshold of others is problematic. Firstly, what is done if these markers aren't present in the panel? Second, different antibodies and their differential location in the panel (in IMC) may yield different signal-to-noise ratios affecting the estimated frequencies of rare populations. Can the authors show quantitatively the effect of signal/noise in these channels has downstream?
11. Please see point 1.
12. Thank you for providing additional metrics.
13. Thank you for the explanation on the differences between clustering methods.
14. Thank you.
15. Thank you for providing the tables with comparisons.

Reviewer #3:

Remarks to the Author:

The authors have done a thorough job of answering all my previous comments, and the manuscript is now significantly improved in terms of supporting the conclusions and general applicability.

One last suggestion is in the reply to the previous major comment #3, Extended Data Figure 7a (Side-by-side comparison of deep-imcyto, Mesmer and Stardist on TRACERx 100 IMC lung cancer data): Compared with Mesmer, some larger cells were generated by deep-imcyto segmentation, see in column 2 and 3. Are those larger cells distinct morphological features or errors caused by segmentation?

Minor Comments:

- Table 1, "Asitr" should be "Astir".

- It would be helpful to have a user guideline for TRACERx-PHLEX on the GitHub page. Currently in the README document, only an overview is available. You have a nice tutorial on <https://tracex-phlex.readthedocs.io/en/main/index.html>, maybe you could put this link also on the GitHub page.

Reviewer #4:

Remarks to the Author:

In general, I commend the authors for considerable revisions and additions which support the clarity of the manuscript and usability of the pipeline. Most of my comments have been explained, and the manuscript edited to clarify, and others addressed with new benchmarking.

Outstanding concerns are present (see below), but would more impact the framing of the tool and its impact rather than the validity of the work. Therefore, I would only suggest mitigating claims regarding the novelty of some pieces of the pipeline, but overall do not question that it performs.

For extended data 4, in order to properly compare expression the cell populations should be in the same order or plots remade to put high and low confidence populations next to each other on the main plot. In this figure different plots have different populations and in different orders. A key aspect here, is not whether low and high confidence cells have different expression levels of the same marker but if there is expression of other markers on a specific cell. This cannot be determined from the figure.

No quantitative comparisons of a cell phenotype were provided for different segmentation methods. Example images are not representative of the overall data quality. Some details on how segmentation impacts the different modules (phenotype, barrier identification) would be useful. This request is to understand the limitations of layering multiple analysis.

Manuscript NCOMMS-23-48349-T

Magness *et al.* - "Deep cell phenotyping and spatial analysis of multiplexed imaging with TRACERx-PHLEX"

We are grateful to the reviewers for their valuable feedback and suggestions, which have strengthened our manuscript. We have incorporated the following modifications after careful consideration of the reviewers' helpful comments:

1. **Systematic benchmarking.** We have now included additional cell phenotyping analyses and compared the performance of TYPEx to all competing approaches, STELLAR, CELESTA and Astir, on the ground truth datasets included in the study for validation.
2. **Impact evaluation of individual steps in multilayered analysis.** We evaluated the impact of the segmentation approach on cell phenotyping and spatial metrics. We performed cell segmentation with 1-, 5- and 10-pixel dilation and compared the cell type results from TYPEx between the different segmentation outputs. We also report how aSMA⁺ fibroblast barrier scores may vary depending on the differences in segmenting the stromal content.
3. **Quantitative evaluation of the impact of noise on cell phenotyping.** Thanks to the reviewer's suggestion, we evaluated the impact of noise in cell phenotyping. We have now assessed quantitatively how the variations in the threshold estimate and the signal-to-noise ratio of a marker can impact the abundance of cell populations. The results show that noise can affect the cell phenotyping results and highlight the need for evaluating the effect of noise systematically by cell phenotyping approaches for imaging mass cytometry data.

In addition to describing the automated, reproducible and open-source computational pipeline for multiplexed imaging analysis, the study now also includes systematic and quantitative evaluation of commonly used analytical tools, thanks to the reviewers' comments throughout the review process. We provide systematic benchmarking of competing approaches, assessment of reproducibility, and quantification of the impact of different factors on performance. In summary, we believe that TRACERx-PHLEX can have a broad use for imaging mass cytometry studies in the growing experimental field.

Any modifications to the text are highlighted in yellow on an annotated version of the manuscript for the reviewers' reference.

Please find the full point-by point response below.

REVIEWER COMMENTS

Reviewer #2 (Remarks to the Author):

The authors have sufficiently addressed my concerns.

2.1. Just one minor comment that Extended Data Figure 4 is cut.

The incorporation of the reviewer's comments throughout the review process helped significantly strengthen our manuscript, and we are grateful for that. We have now modified the layout of **Extended Data Figure 4** (copied below), to ensure its entirety in the revised manuscript.

a

↓
Pathologist Annotations

Binary Masks

↓
TYPEx
Overlay onto deep-imcyto cell mask
Cell Assignment

b

Extended Data Figure 4: Defining binary tissue masks to which cells can be assigned using pathologist annotations. **a**, Pathologist tissue annotation workflow. Pseudo-H&E, H&E and IMC pseudo-coloured composite images are combined into a single file compatible with manual annotation software, such as NDP.view2. Manual annotations such as tumour epithelia and necrosis are converted into binary masks, which are then overlaid onto the cell mask output from deep-imcyto for detailed localisation of TYPEx cell types. **b**, Binary masks of example pathologist annotated features, Alveolar Macrophages and Large Vessels, are shown with accompanying pseudo-coloured composite IMC image with the marker CD68 (Alveolar Macrophages) and cell type map with α SMA (Large Vessels) represented in yellow.

Please see below comments regarding the points raised by reviewer #1:

1.1. Regarding concerns on the generalization of the method beyond TMAs to CODEX data: The analysis presented is a step in the right direction towards benchmarking the methods but it is not done systematically enough. Why was only one patient used in the CODEX HuBMAP dataset (B004, Figure Extended Data Figure 8-9)? Why wasn't precision and recall measured using CELESTA in the CODEX/HuBMAP dataset? The authors should run all competing approaches (including trying Astir again) in both IMC and CODEX datasets, and additional datasets with ground truth.

To address the reviewer's comment and perform systematic benchmarking, we have now included additional analyses for all competing approaches.

We have compared the performance of TYPEx to the available cell phenotyping approaches STELLAR, CELESTA and Astir on the three ground truth datasets included in the study for validation: CRC-CODEX (Schürch *et al.*), BE and HuBMAP. Using the cell subtype definitions input for TYPEx, we have now evaluated the performance of the probabilistic approach for automated cell assignment, Astir, on all validation datasets. For the second phenotyping approach, CELESTA, we had previously included the performance scores for one tissue microarray (TMA) from the Schürch *et al.* dataset (TMA A). The scores were derived from the published confusion matrix in the study by Zhang *et al.* that described this approach. To address the reviewer's suggestions, we have further analysed the images from the second TMA, TMA B, in the Schürch *et al.* dataset. To do this, we used the cell subtype definitions and parameters applied to the TMA A subset and provided with the GitHub distribution of CELESTA. In addition, we applied CELESTA on the two ground truth datasets: BE and HuBMAP B004, using the cell subtype definitions input to TYPEx and formatted according to the CELESTA guidelines. The third approach, STELLAR, is a deep learning approach that requires training data. The performance of STELLAR was previously evaluated on the validation datasets, BE, HuBMAP B004 and Schürch *et al.*. Of note, the performance of STELLAR on the Schürch *et al.* datasets was evaluated on a subset of targeted antigens that overlapped with the training data from a different antibody panel on fresh frozen healthy intestine (regardless of the clone).

For performance evaluation and benchmarking, we chose publicly available datasets for which ground truth data with expert-annotated cell types were available. In the HuBMAP dataset, expert annotations were available for the patient B004 and included eight whole-slide images. As STELLAR requires training data to be run on additional cohorts, the validation datasets were restricted to the three CODEX datasets for which STELLAR training data was available or evaluated.

The results from benchmarking of all four approaches are shown in **Figure 7j** (copied below) and reported in the Results section *Validation and benchmarking of cell phenotyping with TYPEx*. Cell type-specific analyses have now been included in **Figure 7h-i** (F1 score) and **Extended Data Fig. S9c-e** (Precision and Recall).

Based on the evaluation metrics, macro F1-score and mean precision and recall, the performance of TYPEx was comparable to the probabilistic cell phenotyping approaches CELESTA¹ and Astir² and the geometric deep-learning method STELLAR³ (Fig. 7h-j, Extended Data Fig. 9c-e). The macro F1 score on the CRC dataset ranged from 0.5-0.6 for TYPEx, CELESTA and STELLAR and 0.45 for

Astir (Fig. 7j). In this dataset, CELESTA scores on the TMA A derived from a published confusion matrix were higher than those on TMA B derived from our analysis using the same settings for TMA A, suggesting that the optimisation of the cell phenotyping tools affects their performance and may be required for each TMA. In the BE dataset, the macro F1-score, mean precision and recall for TYPEx (0.7-0.74, Fig. 7i, Extended Data Fig. 9d) were comparable to the ones achieved by STELLAR, which incorporates both molecular and spatial information (0.75-0.8) ³. In the HuBMAP dataset, the F1-score for TYPEx of 0.65 was lower than STELLAR (0.8) and generally higher than Astir (0.46) and CELESTA (0.52) (Fig. 7j).

Figure 7j: Comparison of macro F1-scores between TYPEx, CELESTA, Astir and STELLAR on all three validation datasets: CRC-CODEX (Schürch *et al.*), BE, and HuBMAP. WSI = Whole-Slide Image

1.2. Thank you for addressing the concerns regarding computational complexity/time.

1.3. The use of a Z-score in Figure 6b makes the range of the metrics hard to interpret, particularly given that many have nicely bounded ranges. Furthermore, for a few metrics such as "merge", "missed_det_from_merge", "true_det_in_catastrophe", "pred_det_in_catastrophe", "catastrophe", CellPose (nuclear) seems to perform better than deep-imcyto. The authors should mention on the Methods how these metrics are calculated and comment on potential reasons for worst performance.

We thank the reviewer for the suggestion and have now amended **Figure 6b** to remove z-scoring from all metrics (copied in below). To improve display in the heatmap we have normalised bijective cardinality by the total possible number of correct detections in the test dataset. Whether a higher or lower score is desirable for an individual score is now highlighted in the figure legend. By removing the z-score normalisation, CellPose (nuclear) performs better than deep-imcyto by only one metric - "missed_det_from_merge" - and slightly better by "gained detections". However, for all other metrics,

performance of CellPose (nuclear) is comparable to or worse than deep-imcyto. This result suggests that the reviewer recommendation to remove z-score normalisation has helped to alleviate the exaggeration of small absolute differences in segmentation scores between different methods and helped to aid interpretation.

Figure 6: Evaluation of PHLEX segmentation performance compared to standard approaches. - excerpt. b, Heatmap summary of the mean segmentation performance of all metrics shown in **Extended Data Figure 6**. Upper panel shows scores where higher values indicate superior performance. The lower two panels show scores where a lower value is more desirable. *Bijection cardinality was normalised by the total possible number of correct detections in the test dataset.

We have added descriptions for how these metrics were calculated in the section *Segmentation scoring* of the *Methods* -

“All segmentation scoring metrics unless otherwise specified were calculated using the deepcell-toolbox Python library. $n_{\text{predict}}/n_{\text{gt}}$ was calculated as the ratio of predicted objects to the number

of ground truth objects in a given image. Bijective cardinality (the number of labelled objects in the ground truth dataset for which only a single overlapping label in the predicted dataset) and segmentation similarity measure were calculated according to Al-Kofahi *et al.* using a custom Python script.”

1.4. The authors "acknowledge [...] the subjective and time-consuming nature of manual review" involved in the selection of segmentation markers. However, this seems to now only be highlighted in the Tutorial and not in the manuscript. In the current version of the manuscript, TRACERx-PHLEX is still labeled as an "automated" pipeline. Furthermore, the previous reviewer comments regarding "the rationale to choose filtering parameters" not being explained was not incorporated in the manuscript and should be present in the Methods section.

We thank the reviewer for their comments and recognise the need to more clearly highlight the differing levels of user configuration required to apply different modes of deep-imcyto cell segmentation in the main manuscript and Methods.

We have acknowledged in the *Discussion* that deep-imcyto is an automated pipeline; however, some options may require prior manual configuration of the input files by the user. Specifically, use of deep-imcyto in its *CellProfiler* mode requires manual configuration of a segmentation workflow by a user, whilst the *simple* mode can be invoked as is from the command line (default pixel dilation +1) with just basic input/output configuration (e.g. provision of files to be processed, naming of output directories etc).

The *CellProfiler* mode provides the facility for user-extension of a segmentation workflow through the use of a customisable CellProfiler pipeline file. While producing this .cppipe file is a non-automated step with the processes that it could contain chosen by the user, once the user has designed this file, deep-imcyto's *CellProfiler* mode will also execute automatically with no user intervention required.

We used *CellProfiler* mode to run a CellProfiler-based procedure we developed (Multiplexed Consensus Cell Segmentation, M CCS) that contained steps specific to our TRACERx lung dataset and offered greater capture of fibroblasts and irregular cell bodies in the lung microenvironment than deep-imcyto's *simple* mode. Using deep-imcyto's *CellProfiler* mode is an optional step, and whether it is advantageous to use may depend on the type of dataset being processed, but we believe it may prove useful to other users who have to contend with more complex features of their own images. For this they need only basic experience in CellProfiler, but not necessarily the skills to run analyses reproducibly and en masse.

We consider the ability to run a CellProfiler file through deep-imcyto a core functionality of the pipeline. However, the specifics of M CCS are related to segmentation of the TRACERx lung samples. As requested by the reviewer, we now provide an overview of the M CCS approach as implemented on our TRACERx lung samples in the *Methods* section of this manuscript, including rationale for the selection of filtering, overlap and size parameters chosen. We also provide more detailed information of M CCS in the Tutorial should a user wish to mimic this particular segmentation approach. The design and optimisation of M CCS segmentations (or any user-specified CellProfiler pipeline) may well be laborious, and we have amended the *Discussion* to state this as a limitation of deep-imcyto should a user select this particular operational mode.

1.5. Thank you for the explanation regarding scaling of the distributions. Please make this section clearer in the Methods section as well and not only in the Tutorial so the approach can be reproduced outside of the code provided by the authors as well.

We thank the reviewer for the suggestion. We have now clarified in the *Methods* section the scaling factor and the case scenarios when it should be modified.

In the *Methods* section *Major cell lineage and cell subtype definitions*, we included the following paragraphs. To configure CellAssign for analysis of pixel intensities,

The raw marker intensities were multiplied by a factor of ten. The factor 10 was chosen from several evaluated factors (1, 10, 100), by comparing the proportion of CD4 and CD8 major cell lineages in the probabilistic model to the expected T cell proportions CD4⁺ and CD8⁺ in the TRACERx cohort derived from orthogonal flow cytometry datasets. The default value is applicable to raw input cell-by-marker intensity tables and deep-imcyto in simple mode. The default factor of ten was also applied to the z-score normalised intensities in the HuBMAP and BE datasets.

If the input intensities are rescaled in the range 0-1, the argument magnitude in the config file typing params should be used to set the scaling factor. The intensities in the cell-by-marker matrix generated by deep-imcyto using our MCCS procedure in CellProfiler mode needed to be further multiplied by a factor of 10⁵ to be scaled back to the raw pixel intensities. The intensities output by MCCS are scaled in the range from 0-1 due to its CellProfiler implementation. MCCS rescales the intensities of input images for all channels, dividing by a user-defined number exceeding the maximum intensity in the dataset, 10⁵, to ensure that the intensity values fall in the range 0-1 for the two IMC TRACERx datasets. Notably, the intensity distributions remain untransformed. This refactoring is implemented to meet a requirement from CellProfiler that all saved .tiff files output from the full_stack_preprocessing step should have intensity ranges between 0-1. Therefore, for all analysed datasets, the parameter magnitude was set to the recommended default value of 10, except for the TRACERx data analysed with the MCCS procedure, where magnitude was set to 10⁶.

1.6. Thank you for the explanation

1.7. Thank you for detailing the rationale in more detail.

1.8. Thank you for acknowledging the limitations. Please mention them in the manuscript (e.g. Discussion section), and not only in the Tutorial.

We agree with the reviewer and have detailed the limitations in the *Discussion* of the revised manuscript.

With respect to TYPEx, we acknowledge the factors that could affect its performance as follows.

The performance of TYPEx could be affected by high noise levels of markers, the segmentation approach, the validity of the user-provided cell type definitions and the robustness of the clustering approach within a cell lineage and confidence group.

In addition, we acknowledge that, while deep-imcyto is an automated pipeline, certain options may involve manual configuration by the user.

While deep-imcyto is a fully automated pipeline, its use in CellProfiler mode involves prior manual configuration of the input files by the user.

1.9. I appreciate the effort made by the authors to illustrate the effect of threshold decision, however the original reviewer comment requested to "demonstrate this quantitatively". Can the authors for example simulate different levels of noise to real-world datasets and assess the effect variance in threshold estimate has on the abundance of populations/positive populations (e.g. with correlation).

We thank the reviewer for the suggestion and agree that the impact of noise is an important factor in cell phenotyping. We have now assessed quantitatively how the variations in the threshold estimate can impact the abundance of cell populations. The new results have now been included in **Supplementary Figure S7** and are addressed in the *Discussion*.

We simulated different noise levels on marker channels from the panel *T cell & Stroma* in the TRACERx IMC dataset, for which the threshold estimation has been illustrated in **Figure 3b-d**. We simulated ion shot noise by modelling the ion counts as a Poisson process for a given channel, similarly to previous work on noise simulation approaches on imaging mass cytometry data ⁴. The pixel intensity with simulated noise was derived as

$$S_p = P(R_p / \gamma), \{ \gamma = 1 \text{ high noise level}; \gamma = 2 \text{ low noise level} \}$$

where S_p represents the intensities of the pixels p with simulated noise for a given channel, and $P(r)$ is the Poisson noise. R_p corresponds to the raw measured signal for the channel, and γ is the scale factor for adjusting the level of signal-to-noise ratio (SNR). Two noise levels were added to the raw signal: high noise level/low SNR where $\gamma=1$ and low noise level for $\gamma=2$. Hot pixel removal and spillover compensation is performed as a pre-processing step before calculating the pixel intensities; therefore, hot pixel noise and signal spillover were not simulated in addition to ion shot noise.

We explored two scenarios:

1) if the noise levels of the threshold-defining markers (CD3, CD4, CD8a) remain constant and the noise levels of other markers change, how does the abundance of the other markers/cell populations change?

and

2) if the noise levels of the threshold-defining markers change, how does the abundance of all markers/cell populations change?

The latter question has been elaborated on in the reply to the next reviewer's point **(1.10)**.

In relation to the first question, we assessed how the quantification of the respective cell marker positivity calls changes, when we impute noise to the channels CD31 and aSMA, which were not used to define the D-score thresholds (**Supplementary Figure S7c**, copied below). In this scenario, the noise levels of the

threshold-defining markers remained constant, and, as a result, the thresholds for low- and high-confidence populations did not change. Therefore, we applied the same D-score threshold on both the raw pixel intensities and the intensities with simulated noise.

Supplementary Figure S7c-d. Effect of different noise levels on cell positivity calls and cell type counts for two markers: CD31 (c) and aSMA (d) in the *T cells & Stroma* panel. Two levels of noise were simulated: low and high. The proportion of change in CD31⁺/Endothelial cells and aSMA⁺ cells was illustrated relative to the corresponding proportion based on raw intensities without noise simulation (no sim.)

We compared the positive cells for the two markers: CD31 and aSMA cells in the *T cells & Stroma* panel (Supplementary Figure S7c-d). The proportion of CD31⁺ cells with different noise simulation levels remained comparable to proportion based on the raw intensities, with an increase of up to 2.4% at high noise levels. Adding high noise levels to CD31 increased the proportion of Endothelial cells by 1.5% compared to the number of Endothelial cells based on raw CD31 intensities. However, high noise levels resulted in a decrease of 10% in cells positive for aSMA⁺ cells. Similarly, the number of cells annotated with the cell subtype aSMA⁺ cells decreased by 4.6%. Of note, the change in cell subtype annotations is lower than the change in positive cells, because the cell assignment considers the combination of all positive markers. Finally, we compared pairwise the number of Endothelial cells and aSMA+ cells per image between the three measurements no sim, low and high noise levels and observed a strong positive correlation (Spearman correlation coefficient $\rho=1$, $p<2e-16$).

In the revised manuscript, we now discuss that the performance of TYPEx can be affected for markers with low signal-to-noise ratio.

1.10. Thank you for the explanation. However, the strategy to use the intensity range of some markers (CD3, CD4 and CD8a) to determine the threshold of others is problematic. Firstly, what is done if these markers aren't present in the panel? Second, different antibodies and their differential location in the panel (in IMC) may yield different signal-to-noise ratios affecting the estimated frequencies of rare populations. Can the authors show quantitatively the effect of signal/noise in these channels has downstream?

Thanks to the reviewer's suggestion, we have now incorporated this important analysis in the revised manuscript.

Using the noise simulation approach described in point 1.9, we simulated different noise levels for the intensities of the three markers used to estimate the threshold. We considered three simulation scenarios: low noise levels for one of the markers (CD8a low), high noise levels for the same marker (CD8a high), and high noise levels for all three markers (CD3, CD4, CD8a). For each case, we evaluated the thresholds for the low and high confidence groups given the intensities of the defining markers with simulated noise. **Supplementary Figure S7a** shows the changes in the thresholds depending on the noise level and number of defining markers with simulated noise. Adding a low noise level to one of the markers resulted in similar thresholds within two decimal points for the raw and simulated data: 0 and 0.006 for high and low confidence groups, respectively. Adding high noise levels to one the three markers decreased the low confidence threshold by 0.045. Furthermore, high noise levels for all three markers decreased both the low and high confidence thresholds by 0.025 and 0.028, respectively. Therefore, these results demonstrated that the threshold estimate is affected by the noise levels of the defining markers.

Supplementary Figure S7a-b. Impact of different noise levels on the threshold estimate and change in cell population abundances. **a**, Changes in the low and high confidence-group thresholds depending on the noise level (low and high) and markers with simulated noise (CD8a and CD3/CD4/CD8a). **b**, Proportion of cells that changed annotation due to the threshold change and simulated noise.

We next evaluated the change in cell abundance as a result of the threshold variation (**Supplementary Figure S7b**). As a result of the threshold change with the added high noise level to CD8a intensities, 2.4% of the cells changed their cell type annotation. High noise levels for all three markers led to 6% of cells changing their annotation. Overall, we observed a strong correlation of the cell count for all cell types per image in pairwise comparison of the three runs with and without simulations. The minimum Spearman correlation coefficient from the pairwise comparisons was 0.96. In summary, noise and low signal-to-noise ratio can affect the performance of cell phenotyping on the corresponding channels, resulting in reannotation of up to 6% of cells. This observation has now been addressed in the *Discussion* and highlights the need for evaluating the effect of noise systematically by cell phenotyping approaches for imaging mass cytometry data.

We also agree with the reviewer that while CD3, CD4 and CD8a are frequently integral to an antibody panel, these markers may not be included in certain experimental designs. When any of the three T cell markers are not profiled in the user's antibody panel, the implementation of TYPEx allows for any markers with similar patterns of co-expression and mutual exclusivity to be specified in the input typing params config file. In short, TYPEx estimates an optimal D-score threshold that minimises the populations listed as "rare" but maximises those listed "high_frequency". By default, in the config file, we specify that the double-positive CD3⁺CD4⁺CD8a⁺ and single-positive CD3⁻CD4⁺CD8a⁺ cells are expected to be rarely found in peripheral non-lymphoid tissue, whereas CD3⁺CD4⁺ and CD3⁺CD8a⁺ to be the "high_frequency" populations in the analysed cohort. So far, we have tested the combination of CD45, CD4 and CD8 on an in-house dataset, where CD3 was not included in the antibody panel. CD45 can be used to replace CD3, where the frequent subpopulations can be defined as CD45⁺CD4⁺ and CD45⁺CD8⁺ whereas rare subpopulations can be CD4⁺ (CD45⁻), CD8⁺ (CD45⁻), CD4⁺CD8⁺, and CD45⁺CD4⁺CD8⁺. Details on how we configured the input file are available in the TRACERx-PHLEX documentation, section *Marker selection for D-score threshold* under TYPEx/Guide.

1.11. Please see point 1.

1.12. Thank you for providing additional metrics.

1.13. Thank you for the explanation on the differences between clustering methods.

1.14. Thank you.

1.15. Thank you for providing the tables with comparisons.

Reviewer #3 (Remarks to the Author):

The authors have done a thorough job of answering all my previous comments, and the manuscript is now significantly improved in terms of supporting the conclusions and general applicability.

3.1. One last suggestion is in the reply to the previous major comment #3, Extended Data Figure 7a (Side-by-side comparison of deep-imcyto, Mesmer and Stardist on TRACERx 100 IMC lung cancer data): Compared with Mesmer, some larger cells were generated by deep-into segmentation, see in column 2 and 3. Are those larger cells distinct morphological features or errors caused by segmentation?

We are grateful for the reviewer's suggestion. Closely packed and indistinct nuclei are a challenge for any segmentation method. Moreover they are a challenge for human experts when it comes to generating the ground truth annotations, as it is difficult to identify true boundaries by eye. Furthermore, as others have found for other imaging modalities, inter-operator variability of ground truth is an issue for IMC too. Typically, disagreement is more common in noisy and/or close-packed regions of nuclei. We think that the nuclei being referred to probably relate to regions such as this, and so it is difficult to make a definitive call as to whether the large detections of deep-imcyto are a 'true' morphological feature versus Mesmer's predictions. However the prediction will be morphologically plausible, as otherwise it would be flagged

by deep-imcyto's morphological error correction model. While it is possible that deep-imcyto under-segments these particular instances, we were also able to see instances of potential over-segmentation by Mesmer in the regions the reviewer refers to.

Regardless of these small differences between deep-imcyto and leading alternatives, we believe the data and representative images presented show that deep-imcyto is a very capable model for IMC image segmentation that will prove sufficiently accurate for many users.

Minor Comments:

3.2. Table 1, "Asitr" should be "Astir".

We thank the reviewer for highlighting this error, which has now been corrected in the revised manuscript.

3.3. It would be helpful to have a user guideline for TRACERx-PHLEX on the GitHub page. Currently in the README document, only an overview available. You have a nice tutorial on <https://tracex-phlex.readthedocs.io/en/main/index.html>, maybe you could put this link also on GitHub page.

Thanks to the reviewer's suggestion, we have included a link to the TRACERx-PHLEX User Guide on the GitHub page of TRACERx-PHLEX and the GitHub page of each individual module, deep-imcyto, TYPEx, Spatial-PHLEX.

Reviewer #4 (Remarks to the Author):

In general, I commend the authors for considerable revisions and additions which support the clarity of the manuscript and useability of the pipeline. Most of my comments have been explained, and manuscript edited to clarify, and others addressed with new benchmarking.

Outstanding concerns are present (see below), but would more impact the framing of the tool and its impact rather than the validity of the work. Therefore, I would only suggest mitigating claims regarding the novelty of some pieces of the pipeline, but overall do not question that it performs.

4.1. For extended data 4, in order to properly compare expression the cell populations should be in the same order or plots remade to put high and low confidence populations next to each other on the main plot. In this figure different plots have different populations and in different orders. A key aspect here, is not whether low and high confidence cells have different expression levels of the same marker but if there is expression of other markers on a specific cell. This cannot be determined from the figure.

We have incorporated the reviewer's suggestions by amending **Extended Data Figure 4** and including heatmaps of marker intensities across cell subtypes for each of the datasets included in the study.

We apologise for the lack of clarity in the submitted **Extended Data Figure 4**. We had previously included the intensity distribution plots for two antibody panels and, acknowledge that, the alternating order of

markers from the two panels made it difficult to follow the same cell subtypes between the boxplots. **Extended Data Figure 4** now includes only the panel *T cells & Stroma*, where each intensity distribution plot shows the cell subtypes in the same order. As per the reviewer's suggestion, the low and high confidence populations are plotted next to each other. An excerpt from the figure is copied below.

Supplementary Figure 4 - excerpt. Pixel intensity distributions for cell-subtype specific markers across different cell subtypes, split by the confidence group. Each point represents the median raw intensity of cells from a given cell subtype per image. Data from *T cells & Stroma* panel in the TRACERx 100 IMC cohort included.

To illustrate the key aspect underlined by the reviewer and show the expression of different markers on a specific cell subtype, below we further include heatmaps of all markers in the antibody panel for each dataset included in the study (**Reviewer Figure 4.1**). The heatmaps for the TRACERx IMC datasets using the *T cells & Stroma* and *Pan-Immune* panels are included in a manuscript under revision by Enfield *et al.*, where the application of PHLEX revealed biological insights into immune evasion, spatial heterogeneity, tumour progression and clinical outcome (Enfield *et al.*, under review). The heatmaps corresponding to the Barrett's Esophagus dataset and HuBMAP B004 have been included in **Supplementary Figure S5b** and the **Tutorial Figure 4**, respectively. The figure corresponding to the dataset Schürch *et al.* is not included in the manuscript.

Reviewer Figure 4.1. Heatmaps of marker pixel intensities across the cell subtypes annotated by TYPEx. Median pixel intensities per cell subtype were z-score normalised for each of the following datasets: TRACRERx IMC datasets from Enfield *et al.* for panels T cells & Stroma (**a**) and *Pan-Immune* (**b**), Barrett's Esophagus (BE) and tonsil datasets (**c**, **Supplementary Figure S5b**), Schürch *et al.* (**d**), and HuBMAP (**e**, **Tutorial Figure 4**).

4.2. No quantitative comparisons of a cell phenotype were provided for different segmentation methods. Example images are not representative of the overall data quality. Some details on how segmentation impacts the different modules (phenotype, barrier identification) would be useful. This request is to understand the limitations of layering multiple analysis.

We agree with the reviewer and acknowledge the benefit of presenting quantitative comparison of the outputs using different segmentation methods. To address this suggestion, we performed additional analyses and evaluated how the cell phenotype quantification and the barrier scores change depending on the segmentation approach.

To examine how segmentation impacts the detection of cell phenotypes, we performed cell segmentation with dilation of 1px, 5px and 10px, using deep-imcyto in *simple* mode for panel *T cells & Stroma*. Each of the segmentation outputs were then used as input to TYPEx for cell type analysis. The cell phenotyping results from the three dilation runs were then compared to those derived from the Multiplexed Consensus Cell Segmentation (MCCS) approach.

We observed strong positive correlation for all major cell types identified with the panel *T cells & Stroma*, except for *T cells - Other* ($n=102$ cells with MCCS) (**Supplementary Figure S8a**). However, increasing the dilation pixels resulted in a higher proportion of double-positive $CD4^+CD8a^+$ T cells and higher count of cells rendered Ambiguous (**Supplementary Figure S8b-c**). The proportion of double-positive T cells was twice higher in dilation runs with 5 and 10 pixels compared to 1 pixel and MCCS. Furthermore, as the number of dilation pixels increased, the number of identified Endothelial cells decreased. Similarly, the highest number of $aSMA^+$ cells and Endothelial cells were captured with the MCCS approach compared to the dilation approaches. These results confirmed that MCCS segmentation results in higher stromal content and higher number of typed $aSMA^+$ cells and Endothelial cells compared to simple segmentation approaches, with 1-, 5-, and 10-pixel dilations.

a

b

c

Supplementary Figure S8. Comparison of cell phenotyping between different segmentation approaches.

Four segmentation approaches were compared: Multiplexed Consensus Cell Segmentation (MCCS) and cell segmentation with nuclear dilation of 1px, 5px and 10px using deep-imcyto in *simple* mode for the panel *T cells & Stroma*. Each of the segmentation outputs were then used as input to TYPEx for cell type analysis. **a**, Spearman correlation coefficients from pairwise comparison of the segmentation approaches for each major cell lineage. **b**, Comparison of the proportion of double-positive, CD4⁺CD8a⁺ T cells estimated with TYPEx using the four segmentation approaches as input. **c**, Comparison of the number of annotated cell types, Ambiguous, Endothelial cells and aSMA⁺ cells estimated with TYPEx based on the four segmentation approaches.

We also examined how aSMA⁺ fibroblast barrier scores varied when either including or excluding the non-nucleated cells picked up during the MCCS procedure as a means of assessing the impact on conclusions drawn by incorporating these additional cells using our workflow. **Reviewer Figure 4.2 (top panel)** shows a Spearman correlation plot for the aSMA⁺ fibroblast barrier scores calculated across 121 LUAD and LUSC tumour cores from the TRACERx lung dataset when including all cells ('All Cells') or excluding the non-

nucleated cells identified using MCCS (“Nucleated Cells Only”). Whilst individual cores show changes in the scores using the different segmentation approaches, the overall Spearman correlation is very high ($\rho=0.95$, $p<2.2e-16$), suggesting that, in this case, the segmentation approach may be unlikely to change overall conclusions related to barrier score analyses. Nonetheless, individual regions of interest provide compelling examples as to the benefits of including non-nucleated aSMA+ cells in better recapitulating tissue structure (**Reviewer Figure 4.2 (bottom panels)**). In the bottom right panel, non-nucleated cells are represented in white, providing an example of an instance in which peritumoural aSMA+ cells would not be picked up by an approach relying solely on nuclear seeds. Future work should look to further improve capture of this prevalent feature of tumour images.

We have now mentioned in the *Discussion* that the segmentation approach can affect downstream cell phenotyping analyses.

Supplementary Figure S8: d, Spearman correlation for aSMA+ fibroblast barrier scores (adjacent fraction mean score) for 121 LUAD and LUSC tumour cores in the TRACERx lung dataset, calculated using all cells identified using MCCS and only the nucleated cells identified using MCCS. **e**, Cropped area from a LUAD

tumour core as preprocessed IMC composite (FIJI auto brightness/contrast adjustment + despeckling; red - pancytokeratin, green - CD8a, blue - DNA, yellow - aSMA) **(left)** and cell type map (red - Tumour cell, green - CD8 T cell, yellow - aSMA fibroblast, white - non-nucleated cells from MCCS, other colours - other cell types) **(right)**.

References

1. Zhang, W. *et al.* Identification of cell types in multiplexed in situ images by combining protein expression and spatial information using CELESTA. *Nat. Methods* **19**, 759–769 (2022).
 2. Geuenich, M. J. *et al.* Automated assignment of cell identity from single-cell multiplexed imaging and proteomic data. *Cell Syst* **12**, 1173–1186.e5 (2021).
 3. Brbić, M. *et al.* Annotation of spatially resolved single-cell data with STELLAR. *Nat. Methods* **19**, 1411–1418 (2022).
 4. Lu, P. *et al.* IMC-Denoise: a content aware denoising pipeline to enhance Imaging Mass Cytometry. *Nat. Commun.* **14**, 1601 (2023).
-

Reviewers' Comments:

Reviewer #2:

Remarks to the Author:

Thank you to the authors for heeding my recommendations for clarity and transparency in the manuscript, as well as highlighting limitation in the performance of TYPEX (e.g. in markers with low signal-to-noise ratio).

One final note is that STELLAR seems consistently more accurate at cell type classification (Fig 7i-j, ED Fig 9d). I understand that STELLAR is a method that requires training labels and therefore not directly applicable to all datasets (although the authors could've probably tested it out-of-distribution). I am not questioning the usefulness or solidity of imcyto-TYPEX, but the authors don't really discuss this, and simply describe STELLAR as a comparably performant approach. I leave up to the authors' discretion whether they want to instead highlight how approaches combining (sparsely) labeled data (perhaps even across heterogeneous data types) with more principled approaches as developed in this paper could be a way forward for the field.

Reviewer #3:

Remarks to the Author:

The authors have addressed my comments.

Reviewer #4:

Remarks to the Author:

I thank the authors for further clarification and edits and am satisfied with the responses.

Manuscript NCOMMS-23-48349B

Magness et al. - “Deep cell phenotyping and spatial analysis of multiplexed imaging with TRACERx-PHLEX”

We are grateful to all reviewers and greatly appreciate their valuable comments and thoughtful suggestions throughout the entire review process, which has helped significantly strengthen the manuscript.

The revised manuscript now incorporates the final reviewers' suggestions and the editorial requests. Please find the response to reviewers' comments below.

Point-by-point response to reviewer's comments

Reviewer #2 (Remarks to the Author):

Thank you to the authors for heeding my recommendations for clarity and transparency in the manuscript, as well as highlighting limitation in the performance of TYPEx (e.g. in markers with low signal-to-noise ratio).

One final note is that STELLAR seems consistently more accurate at cell type classification (Fig 7i-j, ED Fig 9d). I understand that STELLAR is a method that requires training labels and therefore not directly applicable to all datasets (although the authors could've probably tested it out-of-distribution). I am not questioning the usefulness or solidity of imcyto-TYPEx, but the authors don't really discuss this, and simply describe STELLAR as a comparably performant approach. I leave up to the authors' discretion whether they want to instead highlight how approaches combining (sparsely) labeled data (perhaps even across heterogeneous data types) with more principled approaches as developed in this paper could be a way forward for the field.

We agree with the reviewer's suggestion and have now amended the text to deliver the following points related to STELLAR:

In the Results section *Validation and benchmarking of cell phenotyping with TYPEx*, we note that STELLAR showed the best performance based on the F1-score metrics; however, STELLAR was not trained and tested outside of the original distribution. We also highlight that the performance of all tools including STELLAR can be affected by the user's familiarity with the tool, as shown for CELESTA.

The legend text for **Figure 7** now specifies that the STELLAR performance metrics were previously published (and therefore not tested out-of-distribution).

In the *Discussion*, we now compare the limitations and advantages between the different cell phenotyping tools, including STELLAR, and discuss perspectives of multiplexed imaging analyses thanks to the reviewer's suggestion.

- We now add that while the deep learning-based tool STELLAR demonstrated the highest performance scores among the benchmarked cell phenotyping approaches, the factors that affect these scores, such as the user's expertise, remain to be determined. As these scores were reported previously in STELLAR's study⁸, the performance of STELLAR on independently trained models and datasets outside of its original distribution remain to be evaluated to understand how it is affected by different antibody panels and clones, fixation conditions, image processing and segmentation approaches, and multiplexed imaging modalities.
- We also mention that the user's expertise with a cell phenotyping tool likely affects its performance, as shown for CELESTA. We compared CELESTA between two TMAs comprising matched regions from the same tumours using the same parameters. CELESTA had higher performance scores on the TMAs, for which the parameters were optimised by an expert user. Similarly, for TYPEx, the validity of the user-provided cell type definitions can affect its performance.
- We agree with the reviewer and note that future work should seek to develop tools transferable across heterogeneous datasets and robust to noise. Combining deep learning-based with principled approaches can improve performance and transferability of each individual tool, accelerate and guide the training process and increase (pre-)training data volumes, while facilitating data interpretation and discovery.

We thank the reviewer for their suggestions, which helped ensure systematic benchmarking and expanded our manuscript with new aspects, such as the impact of noise on the cell phenotyping.

Reviewer #2 (Remarks on code availability):

The authors have implemented my suggestions regarding documentation for the codebase including tutorials.

We thank the reviewer for reviewing the code and documentation.

Reviewer #3 (Remarks to the Author):

The authors have addressed my comments.

The reviewer's comments and suggestions have been valuable, for example, for improving the benchmarking, validation and usability of our pipeline, for which we are grateful.

Reviewer #4 (Remarks to the Author):

I thank the authors for further clarification and edits and am satisfied with the responses.

We greatly appreciate the reviewer's comments and suggestions, which have expanded our manuscript with new aspects, such as examining the impact of multi-layered analysis on performance.

Reviewer #4 (Remarks on code availability):

Previous versions were installed and run using provided resources, but detailed outputs for each module were not fully examined.

We thank the reviewer for testing the pipeline.